# Sample-efficient Bayesian Optimisation Using Known Invariances

**Theodore Brown**[*1,2]      **Alexandru Cioba**[*3]      **Ilija Bogunovic**[1]

[*]Equal contribution

[1]University College London   [2]United Kingdom Atomic Energy Authority   [3]MediaTek Research

{theodore.brown.23, i.bogunovic}@ucl.ac.uk

alexandru.cioba@mtkresearch.com

## Abstract

Bayesian optimisation (BO) is a powerful framework for global optimisation of costly functions, using predictions from Gaussian process models (GPs). In this work, we apply BO to functions that exhibit *invariance* to a known group of transformations. We show that vanilla and constrained BO algorithms are inefficient when optimising such invariant objectives, and provide a method for incorporating group invariances into the kernel of the GP to produce *invariance-aware* algorithms that achieve significant improvements in sample efficiency. We derive a bound on the maximum information gain of these invariant kernels, and provide novel upper and lower bounds on the number of observations required for invariance-aware BO algorithms to achieve $\epsilon$-optimality. We demonstrate our method's improved performance on a range of synthetic invariant and quasi-invariant functions. We also apply our method in the case where only some of the invariance is incorporated into the kernel, and find that these kernels achieve similar gains in sample efficiency at significantly reduced computational cost. Finally, we use invariant BO to design a current drive system for a nuclear fusion reactor, finding a high-performance solution where non-invariant methods failed.

## 1   Introduction

Bayesian optimisation (BO) has been applied to many global optimisation tasks in science and engineering, such as improving the performance of web servers [30], reducing the losses in particle accelerators [26], or maximising the potency of a drug [1]. The most challenging real-world tasks tend to have large or continuous input domains, as well as target functions that are costly to evaluate.

One particular example of interest is the design and operation of nuclear fusion reactors, which promise to deliver huge amounts of energy with zero greenhouse gas emissions. However, the development of high-performance fusion power plant scenarios is a difficult optimisation task, involving expensive simulations and physical experiments [39, 11]. For tasks like this, it is desirable to develop *sample efficient* algorithms that only require a small number of evaluations to find the optimal solution.

Crucially, the physical world is characterized by underlying structures that can be leveraged to significantly improve sample efficiency. The concept of *invariance*, which describes properties that remain constant under various transformations, is one of the most fundamental structures in nature. A prime example of invariance is found in image classification tasks: a classifier should consistently identify a cat as a cat, regardless of its orientation, position, or size. The central question we answer is the extent to which the sample efficiency of BO can be improved by exploiting these invariances.

We study a general approach for incorporating invariances into any BO algorithm. We represent invariant objective functions as elements of a reproducing kernel Hilbert space (RKHS), and produce invariance-aware versions of existing Gaussian process (GP) bandit optimisation algorithms by using the kernel of this RKHS. We apply these algorithms to synthetic and nuclear fusion optimisation

38th Conference on Neural Information Processing Systems (NeurIPS 2024).

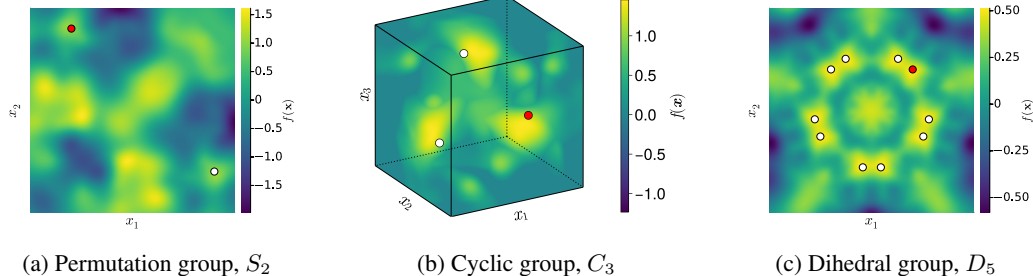


(a) Permutation group, $S_2$     (b) Cyclic group, $C_3$     (c) Dihedral group, $D_5$


Figure 1: Examples of group-invariant functions, sampled from a Gaussian process with the corresponding invariant kernel. Note that observing a point $x$ (**red**) provides information about transformed locations $\{\sigma(x) : \sigma \in G\}$ (white).

tasks, and show that incorporating the underlying invariance into the kernel significantly improves the sample efficiency. We also investigate the behaviour of the algorithms when the full invariant kernel is not known, and provide an empirical study of the performance of invariance-aware BO on target functions that are 'quasi-invariant' (modelled as a sum of invariant and non-invariant components). Our main contribution is the derivation of novel upper and lower bounds on the regret achieved by invariance-aware BO algorithms.

## 1.1 Related work

**Invariance in deep learning.** The role of invariance in deep learning has been extensively explored in the literature. Invariance, particularly to geometric transformations like rotation, scaling, and translation, is crucial for the robust performance of deep learning models. Many deep learning methods incorporate invariance via *data augmentation*, in which additional training examples are generated by transforming the input data [28, 29]. However, the cost of kernelised learning methods often scales poorly with the number of training points $n$ – Gaussian process inference, for example, incurs $\mathcal{O}(n^3)$ time and $\mathcal{O}(n^2)$ memory cost [47] – which means that data augmentation is prohibitively expensive. In this work, we adopt an alternative approach by directly embedding invariance properties into the model's structure, which effectively reduces computational complexity [44, 15].

**Structure-informed BO.** There is an extensive body of empirical BO literature that exploits known patterns present in the behaviour of a system. In [24], BO is applied after a variety of physics-informed feature transformations to the state; in [48] the authors employ BO to handle optimization over crystal configurations with symmetries found using [40]. However, in these works, only empirical results are given and asymptotic rates on sample complexity are not estimated.

**Invariant kernel methods.** The theory of group invariant kernels was introduced in [18, 27]. Using these ideas, [46] developed efficient variational methods for learning hyperparameters in invariant Gaussian processes. Invariant kernel ridge regression (KRR) for finite groups has been studied for translation- and permutation-invariant kernels [33, 4]. [38] quantify the gain in sample complexity for kernels invariant to the actions of Lie groups on arbitrary manifolds and tie the sample complexity bounds to the dimensions of the group. Motivated by the invariance properties of convolutional neural networks, [27, 33, 4] focus on kernelised *regression* with the Neural Tangent Kernel; we instead consider the kernelised *bandit optimisation* setting with the more general Matérn family of kernels. These methods can be readily extended to less strict notions of so-called *quasi*-invariance [18, 46, 4].

**Matérn kernels on manifolds.** The previous studies of invariant kernel algorithms rely on the spectral properties of kernels on the hypersphere [33, 4]; the relevant spectral theory for Matérn kernels on Riemannian manifolds (including the hypersphere) was developed in [9]. These geometry-aware kernels were applied to optimisation tasks in [20]; however, the structure they include is restricted to the geometry of the domain only, rather than invariance to a set of transformations.

**Regret bounds for Bayesian optimisation.** The techniques for upper-bounding the regret of Bayesian optimisation algorithms using the kernel-dependent quantity known as maximum information gain were developed in [37]. Subsequent studies [43, 8, 14, 36, 19, 41] have focused on improving regret bounds in both cumulative and simple regret settings. [22] provided an upper bound for regret in BO using the convolutional NTK, which exhibits invariance only to the cyclic

group. [23] explored graph neural network bandits, offering maximum information gain rates by embedding permutation invariance using additive kernels. In [25], a permutation-invariant kernel is constructed in the same way as ours, and a regret analysis provided. Our work expands upon these approaches by considering more general (non-additive) kernels and broader transformation groups, demonstrating that the regret bounds from [22] can be shown to be a specialisation of ours. Lower bounds for kernelised bandits, including their robust variants ([7, 6]), have been examined in [35] and [12]. In this paper, we derive novel lower bounds on regret for the setting with invariant kernels.

## 1.2 Contributions

Our main contributions are:

- We develop new upper bounds on sample complexity in Bayesian optimisation with invariant kernels, which explicitly quantify the gain in sample complexity due to the number of symmetries. We extend several results from the literature, as our statement applies to a wide class of groups. This is, to our knowledge, the first time such bounds are given in this degree of generality.
- We show how results from the literature (e.g. [12]) can be extended to obtain lower bounds on the sample complexity of invariant BO, using a novel construction and quantification of members of the RKHS of invariant functions. We extend the methods in the literature to the hyperspherical domain, giving a blueprint for similar constructions on other Riemannian manifolds.
- We conduct experiments that support our theoretical findings. We demonstrate that incorporating invariance to some *subgroup* of transformations is sufficient to achieve good performance, a result of key importance when the full invariance is not known or is too expensive to compute. We also demonstrate the robustness of the invariant kernel method, in the case where the target function deviates from true invariance.
- We apply these methods to solve a real-world design problem from nuclear fusion engineering that is known to exhibit invariance; our invariance-aware method finds better solutions in fewer samples compared to standard approaches.

## 2 Problem statement

We begin by providing an overview of invariant function spaces. Consider a finite subgroup of isometries, $G$, on a compact manifold, $\mathcal{X}$, embedded in $\mathbb{R}^d$. A function $f : \mathcal{X} \to \mathbb{R}$ is *invariant* to the action of $G$ if, for all $\sigma \in G$,

$$f \circ \sigma = f \quad \text{i.e.} \quad f(x) = f(\sigma(x)) \quad \forall x \in \mathcal{X}. \tag{1}$$

We provide some examples of invariant functions for different groups $G$ in Figure 1.

In this paper, we study the optimisation of invariant functions that belong to a reproducing kernel Hilbert space (RKHS). The invariance of the RKHS elements translates to invariance properties of the kernel. For $k : \mathcal{X} \times \mathcal{X} \to \mathbb{R}$, we say that $k$ is *simultaneously invariant* to the action of $G$ if, for all $\sigma \in G$ and $x, y \in \mathcal{X}$,

$$k(\sigma(x), \sigma(y)) = k(x, y), \tag{2}$$

and is *totally invariant* to the action of $G$ if, for all $\sigma, \tau \in G$ and $x, y \in \mathcal{X}$,

$$k(\sigma(x), \tau(y)) = k(x, y), \tag{3}$$

as introduced in [18]. Note that both inner product kernels $k(x, y) = \kappa(\langle x, y \rangle)$ and stationary kernels $k(x, y) = \kappa(\|x - y\|)$ satisfy simultaneous invariance, as $G$ consists of isometries (which preserve distances and inner products).

Next, we outline the relationship between the invariance of the kernel and the invariance of the functions in the associated RKHS. In particular, the following proposition (Proposition 1) identifies the space of invariant functions as a subspace of the RKHS of a simultaneously invariant kernel. Moreover, it turns out that this subspace is an RKHS in its own right, with a corresponding kernel that is *totally invariant*. For a proof of these properties, see Appendix A.1.

**Proposition 1** (RKHS of invariant functions). *Let $\mathcal{X}$ and $G$ be as defined above. Consider a positive definite and simultaneously invariant kernel $k : \mathcal{X} \times \mathcal{X} \to \mathbb{R}$. The kernel defines a reproducing kernel Hilbert space $\mathcal{H}_k$, whose elements are functions $f : \mathcal{X} \to \mathbb{R}$, and a corresponding Hilbert space norm $\| \cdot \|_{\mathcal{H}_k}$. Define the symmetrization operator $S_G : \mathcal{H}_k \to \mathcal{H}_k$, such that*

$$S_G(f) = \frac{1}{|G|} \sum_{\sigma \in G} f \circ \sigma. \tag{4}$$

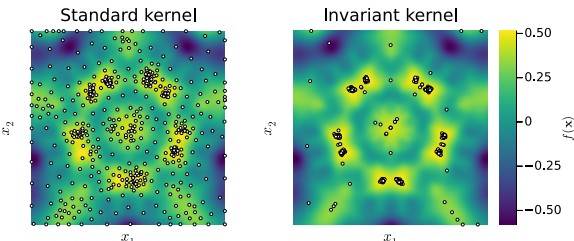

Figure 2: UCB run with standard $k$ and invariant kernel $k_G$ applied to an invariant objective; queried points in white. UCB with the invariant kernel requires significantly fewer samples to find the optimum.

*Then, $S_G$ is a self-adjoint projection operator, whose image $\mathrm{Im}[S_G]$ is the subspace of $\mathcal{H}_k$ that contains G-invariant functions. Moreoever, $\mathrm{Im}[S_G]$ is an RKHS in its own right, $\mathcal{H}_{k_G}$. For $x, y \in \mathcal{X}$, the reproducing kernel of $\mathcal{H}_{k_G}$ is*

$$k_G(x, y) = \frac{1}{|G|} \sum_{\sigma \in G} k(\sigma(x), y). \tag{5}$$

We consider a kernelised optimisation setup with bandit feedback, in which the goal is to find the maximum of an unknown function $f : \mathcal{X} \to \mathbb{R}$ that is invariant to a known group of transformations $G$. We model $f$ as belonging to the invariant RKHS $\mathcal{H}_{k_G}$ from Proposition 1. Without further smoothness assumptions, this task is intractable; similarly to the common non-invariant setting, we make the regularity assumption that $f$ has bounded RKHS norm, $\|f\|_{\mathcal{H}_{k_G}} < B$. The learner is given $k$ and $G$. In Section 3.2, we explore setting $k$ in $k_G$ to common kernels, such as those from the Matérn family.

At each round $t$, the learner selects a point $x_t \in \mathcal{X}$ and receives a noisy function observation

$$y_t = f(x_t) + \eta_t, \tag{6}$$

where $\eta_t$ is drawn independently from a $\sigma_n$-sub-Gaussian noise distribution. The agent reports a point $x_t^*$ and incurs a corresponding *simple regret*, defined as

$$r_t = \arg\max_{x \in \mathcal{X}} f(x) - f(x_t^*). \tag{7}$$

After $T$ rounds, the *cumulative regret* is given by

$$R_T = \sum_{t=1}^{T} \left( \arg\max_{x \in \mathcal{X}} f(x) - f(x_t^*) \right). \tag{8}$$

In the next section, we quantify the impact of invariance on the number of function samples required to achieve a given regret, known as the *sample complexity*.

# 3 Sample complexity bounds for invariance-aware Bayesian optimisation

The bandit problem outlined in the previous section can be tackled using *Bayesian optimisation* (BO) [17]. Intuitively, we expect BO algorithms that utilize the $G$-invariance of the target function to achieve improvements in sample complexity, as each observed $x$ simultaneously provides information about additional transformed locations $\sigma(x)$ for each $\sigma \in G$ (see Figure 1).

In this section, we quantify the performance improvements by deriving bounds on the sample complexity of BO with totally invariant kernels (Equation (3)). We begin with an outline of the BO framework.

## 3.1 Bayesian optimisation of invariant functions

In Bayesian optimisation, predictions from a Gaussian process (GP) are used in conjunction with an acquisition function to select points to query [17]. After collecting a sequence of input-observation pairs, $\mathcal{D} = \{(x_1, y_1), \ldots, (x_t, y_t)\}$, the learner models the invariant target function $f \in \mathcal{H}_{k_G}$ using a GP with a totally invariant kernel, $\mathcal{GP}(0, k_G)$. The posterior predictive distribution of the function value at a specified input location $x$ will be Gaussian with mean and variance given by

$$\mu_t(x) = \boldsymbol{k}_G^T (\boldsymbol{K}_G + \lambda \boldsymbol{I})^{-1} \boldsymbol{y} \tag{9}$$

$$\sigma_t^2(x) = k_G(x, x) - \boldsymbol{k}_G^T (\boldsymbol{K}_G + \lambda \boldsymbol{I})^{-1} \boldsymbol{k}_G, \tag{10}$$

| **Algorithm 1** Max. variance reduction [42] | **Algorithm 2** Upper confidence bound [37] |
|---|---|
| **Input:** $k_G, \mathcal{X}$ | **Input:** $k_G, \mathcal{X}$ |
| 1: **initialize** $\mathcal{D} = \{\}$ | 1: **initialize** $\mathcal{D} = \{\}$ |
| 2: **for** $t = 1, \ldots, T$ **do** | 2: **for** $t = 1, \ldots, T$ **do** |
| 3:     Select $x_t = \arg\max_{x \in \mathcal{X}} \sigma_{t-1}(x)$ | 3:     Select $x_t = \arg\max_{x \in \mathcal{X}} \mu_{t-1}(x) + \beta\sigma_{t-1}(x)$ |
| 4:     Observe $y_t = f(x_t) + \eta_t$ | 4:     Observe $y_t = f(x_t) + \eta_t$ |
| 5:     Append $(x_t, y_t)$ to $\mathcal{D}$ | 5:     Append $(x_t, y_t)$ to $\mathcal{D}$ |
| 6:     Update $\mu_t, \sigma_t$ from Eqs. 9, 10 | 6:     Update $\mu_t, \sigma_t$ from Eqs. 9, 10 |
| 7: **end for** | 7: **end for** |
| 8: **return** $\hat{x}_{\text{opt}}^{\text{MVR}} = \arg\max_{x \in \mathcal{X}} \mu_T(x)$ | 8: **return** $\hat{x}_{\text{opt}}^{\text{UCB}} = x_T$ |

where $\boldsymbol{k}_G = [k_G(x, x_1), \ldots, k_G(x, x_t)]^T \in \mathbb{R}^{t \times 1}$, $\boldsymbol{K}_G$ is the kernel matrix with elements $[\boldsymbol{K}_G]_{i,j} = k_G(x_i, x_j) \, \forall i, j \in 1, \ldots, t$, $\boldsymbol{y} = [y_1, \ldots, y_t]^T$, and $\lambda$ is a regularising term. Note that as this GP has a totally $G$-invariant kernel, $k_G$, it can be interpreted as a distribution over $G$-invariant functions.

We consider two common acquisition functions. The tightest simple regret bounds in the literature are for the *Maximum Variance Reduction* algorithm (MVR), in which the learner iteratively queries the points with highest uncertainty before reporting the point with highest posterior mean [42]. Another algorithm that has been widely studied is *Upper Confidence Bound* (UCB), which balances exploration and exploitation by behaving optimistically in the face of uncertainty [37]. Pseudocode for MVR and UCB is provided in Algorithm 1 and Algorithm 2, respectively. Our analysis is framed in terms of MVR and simple regret, although in Section 4.1 we also investigate empirically the cumulative regret performance of UCB.

A popular family of kernels used in BO is the Matérn kernel. On $\mathbb{R}^d$, the Matérn kernel with parameters $l, \nu$ is given by

$$k(x, x') = \frac{2^{1-\nu}}{\Gamma(\nu)} \left( \frac{\sqrt{2\nu}}{l} \|x - x'\| \right)^\nu K_\nu \left( \frac{\sqrt{2\nu}}{l} \|x - x'\| \right), \tag{11}$$

where $K_\nu$ is the modified Bessel function of the second kind. The parameter $\nu$ controls the differentiability of the corresponding GP, which enables Matérn kernels to define a diverse range of function spaces. As the Matérn kernel $k$ is simultaneously invariant (Equation (2)), we can construct a totally invariant $k_G$ and corresponding RKHS following Proposition 1. The bounds that we derive both hold for the Matérn family; our upper bound also extends to *any* kernel that exhibits polynomial eigendecay.

In Figure 2, we provide a visual example of the impact of using $k_G$ in Bayesian optimisation. We run UCB on a group-invariant objective with the standard and invariant kernels. With the non-invariant kernel $k$, the algorithm spends many samples querying suboptimal points with similar objective values. In contrast, using the invariant kernel $k_G$ allows the algorithm to identify suboptimal regions across the parameter space with only a handful of samples, resulting in significantly faster convergence.

### 3.2   Upper bounds on sample complexity

In this section, let $\mathcal{X} = \mathbb{S}^{d-1} \subset \mathbb{R}^d$ and let $G$ be a finite subgroup of the orthogonal group, $O(d)$, equipped with its natural action on $\mathcal{X}$. Let $k$ be a simultaneously invariant kernel on $\mathbb{S}^{d-1}$, and assume that the eigenvalues of the associated integral operator decay at a polynomial rate $\mu_k = \mathcal{O}(k^{-\beta_p^*})$, for some constant $\beta_p^* > 1$. Many common kernels exhibit this property, such as the standard Matérn kernel on $\mathbb{R}^d$. For GPs defined intrinsically on Riemannian manifolds [45], the corresponding Matérn kernel also exhibits polynomial eigendecay [9]. We impose the following assumption on the spectra of elements in the group $G$, which is in line with the assumptions and spectral decay rates in [4].

**Assumption 1.** *All elements $\sigma \in G$ satisfy the following spectral property:*

$$m_\lambda < d - 4 \quad if \quad \lambda \neq \pm 1, \tag{12}$$

*for all $\lambda \in \text{Spec}(\sigma)$, where $m_\lambda$ is the multiplicity of $\lambda$.*

This assumption excludes certain groups in low dimensions, but we expect it can be relaxed through tighter analysis. In fact, our experiments for $d = 2, 3$ show it is essential as a theoretical artefact only.

Nevertheless, Equation (12) holds for all finite orthogonal groups in dimension $d \geq 10$ (see Lemma 1) as well as for many groups of lower dimension; for example, groups for which $m_\lambda = 1$, so that only distinct eigenvalues are present in the spectra. Under this assumption, we have the following upper bound on sample complexity.

**Theorem 1** (Upper bound on sample complexity of invariance-aware BO). *Let $k$ be a polynomial eigendecay kernel on $\mathbb{S}^{d-1}$ which is simultaneously invariant w.r.t. the orthogonal group $O(d)$, and let $G \leq O(d)$ be a subgroup of $O(d)$ satisfying Assumption 1. Let $k_G$ be the totally invariant kernel, as defined in Proposition 1. Consider noisy bandit optimisation of $f \in \mathcal{H}_{k_G}$, where $\|f\|_{\mathcal{H}_{k_G}} < B$ and $f : \mathbb{S}^{d-1} \to \mathbb{R}$. Then, the maximum information gain (MIG) for the totally invariant kernel after $T$ rounds, $\gamma_T^G$, is bounded by*

$$\gamma_T^G = \tilde{\mathcal{O}}\left( \frac{1}{|G|} T^{\frac{d-1}{\beta_p^*}} \right), \tag{13}$$

*where $\tilde{\mathcal{O}}(\cdot)$ hides logarithmic factors.*

*Moreover, for the MVR algorithm with invariant kernel $k_G$ to achieve expected simple regret $\mathbb{E}[r_T] < \epsilon$, it must hold that*

$$T = \tilde{\mathcal{O}}\left( \left( \frac{1}{|G|} \right)^{\frac{\beta_p^*}{\beta_p^* - d + 1}} \epsilon^{-\frac{2\beta_p^*}{\beta_p^* - d + 1}} \right). \tag{14}$$

*For the Matérn kernel, this corresponds to*

$$T = \tilde{\mathcal{O}}\left( \left( \frac{1}{|G|} \right)^{\frac{2\nu + d - 1}{2\nu}} \epsilon^{-\frac{2\nu + d - 1}{\nu}} \right). \tag{15}$$

*Proof.* See Appendix A.2. ▢

Our upper bound demonstrates that incorporating the group invariance into the kernel is guaranteed to result in sample efficiency improvements that increase with the size of the group, due to the $\frac{1}{|G|}$ factor. Our result for the MIG $\gamma$ of invariant kernels is of independent interest, as it is applicable to a wide range of scenarios. Notably, the linear $\frac{1}{|G|}$ improvement in the information gain from Equation (13) resembles the linear $\frac{1}{|G|}$ improvement in generalisation error in the invariant kernel ridge regression setting [3]. Intuitively, this reflects how the MIG criterion is determined solely by the ability of the kernel to approximate the objective function, and is agnostic to the algorithm used in optimisation.

In our proof, we begin with kernels on $\mathbb{S}^{d-1}$ that are simultaneously invariant w.r.t. the full orthogonal group $O(d)$, and use Proposition 1 to construct kernels that totally invariant kernels w.r.t finite subgroups $G \leq O(d)$. Simultaneously $O(d)$-invariant kernels on $\mathbb{S}^{d-1}$ are in fact dot-product kernels, which allows us to use standard versions of Mercer's theorem in terms of known spectral decompositions. If an alternative decomposition is known for a particular $k$, then it is sufficient that $k$ instead satisfies the weaker condition of simultaneous invariance w.r.t. $G$ only.

### 3.3 Lower bounds on sample complexity

In the next theorem, we present a lower bound on sample complexity which nearly matches the upper bound in its dependence on $|G|$.

**Theorem 2** (Lower bound on sample complexity of invariance-aware BO with Matérn kernels). *Let $\mathcal{X}$ be one of $[0,1]^d$ or $\mathbb{S}^d$. Let $k$ be the standard Matérn kernel on $\mathbb{R}^d$ with smoothness $\nu$, and $G$ be a finite group of isometries acting on $\mathcal{X}$. Let $k_G$ be the totally invariant kernel constructed from $k$ and $G$ according to Proposition 1. Consider noisy bandit optimisation of $f \in \mathcal{H}_{k_G}$, $f : \mathcal{X} \to \mathbb{R}$, where the observations are corrupted by $\sigma$-sub-Gaussian noise. Then, for sufficiently small $\frac{\epsilon}{B}$, there exists a function family $\bar{\mathcal{F}}_\mathcal{X}(\epsilon, B)$ such that the number of samples required for an algorithm to achieve simple regret less than $\epsilon$ for all functions in $\bar{\mathcal{F}}_\mathcal{X}$ with probability at least $1 - \delta$ is bounded by:*

$$T = \Omega\left( \left( \frac{1}{|G|} \right)^{\frac{\nu + d - 1}{\nu}} \epsilon^{-\frac{2\nu + d - 1}{\nu}} \sigma^2 B^{\frac{d-1}{\nu}} \log \frac{1}{\delta} \right), \tag{16}$$

*Proof.* See Appendix A.3. ▢

Our lower bound guarantees that algorithms that are run for a number of sample less than $T$ fail to be $\epsilon$-optimal with high probability. At fixed $T$, there is a trade-off between this probability and the order of the group. However, when combined with the upper bound our results imply that as the order of the group increases, the probability that an algorithm is $\epsilon$-optimal after a fixed number of iterations also increases.

We recover the previously known lower and upper bounds on sample complexity for non-invariant optimisation from [12] and [42] respectively by setting $|G| = 1$. We also note that there is a gap between our bounds, as the exponent of $\frac{1}{|G|}$ in Equation (15) is $1 + \frac{d-1}{\nu}$ but is $1 + \frac{d-1}{2\nu}$ in Equation (16). A likely reason for this gap is in the lower bound, which uses a packing argument for the support sets of functions in the RKHS. While in the absence of the group action, the lower bound and upper bound agree, it is not clear a priori how to achieve the tightest packing of invariant functions.

To achieve the largest improvement in sample complexity, one would like to choose the largest possible $G$ when constructing $k_G$. However, as we will see in our experimental section, there is a computational tradeoff between the sample complexity and the computation of $S_G$, as in the case of very large groups the sums over the group elements becomes a computational bottleneck.

## 4 Experiments

We demonstrate the performance gains from incorporating known invariances in a number of synthetic and real-world optimisation tasks. Previous work has demonstrated the sample efficiency improvements from applying invariance-aware algorithms to regression tasks [3, 33, 38]; our experiments complement the literature by observing the corresponding improvements in optimisation tasks. Code for our experiments and implementations of invariant kernels can be found on GitHub[1].

### 4.1 Synthetic experiments

For each of our synthetic experiments, our goal is to find the maximum of a function drawn from an RKHS with a group-invariant isotropic Matérn-$5/2$ kernel. The groups we consider act by permutation of the coordinate axes, and therefore include reflections and discrete rotations. For `PermInv-2D` (Figure 1a) and `PermInv-6D`, the kernel of the GP is invariant to the full permutation group which acts on $\mathbb{R}^2$ and $\mathbb{R}^6$ respectively by permuting the coordinates. For `CyclInv-3D` it is invariant to the cyclic group acting on $\mathbb{R}^3$ by permuting the axes cyclically (Figure 1b). We provide the learner with the true kernel and observation noise variance to eliminate the effect of online hyperparameter learning. For more detail on our experiments, see Appendix B.

In Figure 3, we show that the algorithm's performance on invariant functions is improved by using an invariant kernel. In all cases, the performance of the invariance-aware algorithm significantly outperforms the baselines, converging to the optimum with many fewer samples. Notably, in the `PermInv-6D` task, the baseline algorithms fail to find the true optimum, whereas the permutation-invariant versions do so without difficulty. We also note that the performance improvement increases with increasing dimension and group size, as predicted by our regret bounds.

For the `PermInv` tasks, we provide an additional baseline: constrained Bayesian optimisation (CBO, [16]). In this case, we constrain the search space of the acquisition function optimiser to the fundamental region of the group's action. For arbitrary groups, finding an analytical formula for this region can be difficult, particularly when the dimension of the domain is high and the group is complicated; however, it is straightforward in the case of the full permutation group. We find that constrained BO performs worse than our method, and in high dimensions is identical to standard (non-invariant) BO. Intuitively, this matches our expectations, as in CBO the kernel is not aware of all of the structure that the RKHS elements possess – that is, while the *domain of exploration* is restricted, the *function class* is not. Observations in CBO contribute no information across the boundaries of the region, unlike in the invariant kernel case, and so the performance improvement of CBO is upper bounded by $\frac{1}{|G|}$.

For the `PermInv-6D` task, we also consider the case where the full invariance is not fully known to the learner; that is, the kernel is not totally invariant w.r.t. the full group, but only a *subgroup*. Our results show that incorporating any amount of the true invariance results in performance improvements. In this case, the true function is invariant to the full permutation group in 6D (720 elements). We evaluate

---

[1] github.com/theo-brown/invariantkernels and github.com/theo-brown/bayesopt_with_invariances

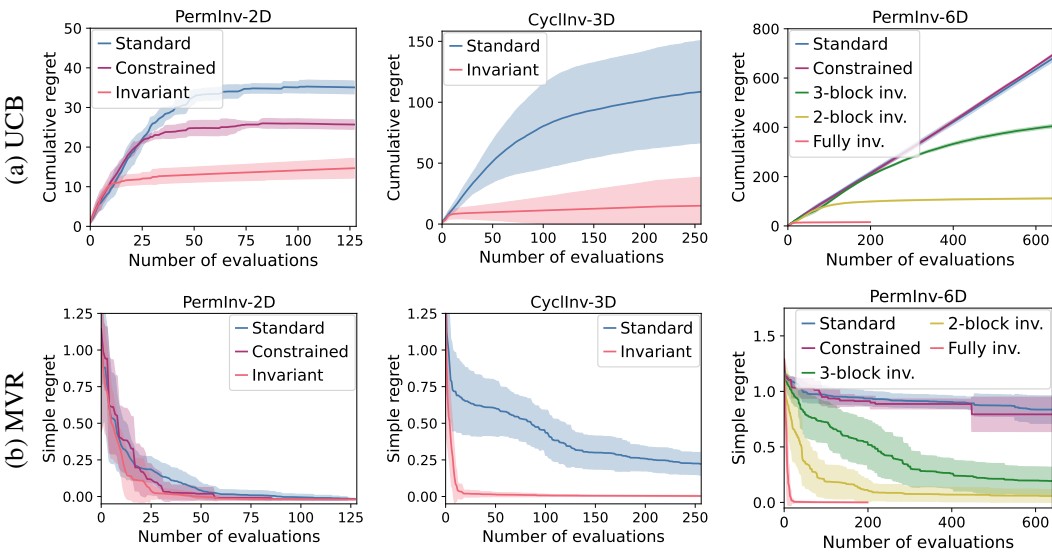

Figure 3: Regret performance of invariant UCB and MVR algorithms across 3 different tasks; lower is better. Non-invariant kernels (**blue**) are outperformed by the full group invariant kernel (**red**) as well as partially specified (subgroup-invariant) kernels (**green** and **yellow**). For the permutation invariant function, the search space of standard BO can be constrained by the fundamental domain of the group (**purple**), but this performs worse than the invariant kernel. Mean ± s.d., 32 seeds.

the performance of the full invariant kernel, the non-invariant kernel, and two subgroup invariant kernels. The subgroups we consider consist of 'block' permutations, which involve reordering the coordinate indices in chunks of length 2 and 3 (leading to groups with 6 and 2 elements respectively). For both algorithms, we see that all kernels that incorporate invariance outperform the non-invariant kernel, even when the difference in the group size considered is large.

## 4.2 Extension: quasi-invariance using additive kernels

In real-world scenarios, it is possible that the underlying function only exhibits *approximate* symmetry. For example, in the fusion application from Section 4.3, the launchers may have slightly different properties and are no longer interchangeable. A 2D example of this 'quasi-invariance' is given in the first column of Figure 4. Although the large-scale features (such as the layout of the peaks) remain approximately symmetric, invariance is not maintained in the detail (the values and shapes of the peaks).

We model quasi-invariance by introducing an additive non-invariant disturbance to the target function. The function can then be considered as belonging to the RKHS $\mathcal{H}_{(1-\varepsilon)k_G + \varepsilon k'}$, where $k_G$ is a $G$-invariant kernel, $k'$ is a non-invariant kernel, and $\varepsilon$ sets the degree of deviation from invariance. In our experiments, we observe that performing BO with the fully invariant kernel still results in significant performance improvements over the non-invariant kernel when $\varepsilon$ is small, achieving performance comparable to BO with the kernel of the function's RKHS. As this is a kind of model misspecification, we refer the reader to [5] for a theoretical discussion on the performance of BO with misspecified kernels.

## 4.3 Real-world application: nuclear fusion scenario design

In this section, we use invariance-aware BO to efficiently solve a design task from fusion engineering.

The leading concept for a controlled fusion power plant is the *tokamak*, a device that confines a high-temperature plasma using powerful magnetic fields. Finding steady-state configurations for tokamak plasma that are simultaneously robustly stable and high-performance is a challenging task. Moreover, evaluating the plasma operating point requires high-fidelity multi-physics simulations, which take several hours to evaluate one configuration due to the complexity and characteristic timescales of the physics involved [34]. Iterating on a design requires many such simulations, which has led to an increased focus on highly sample-efficient algorithms for tokamak design optimisation [13, 32, 10, 31].

We consider a scalarised version of the multi-objective optimisation task presented in [10], in which the goal is to shape the output of a plasma current actuator to maximise various performance and ro-

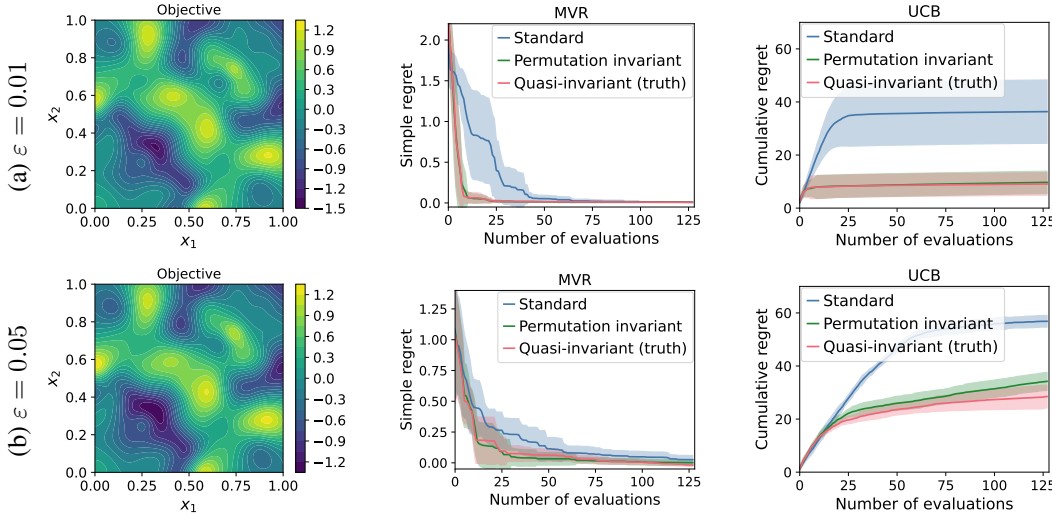

Figure 4: Performance of MVR and UCB on *quasi-invariant* functions. Regret shown for the noninvariant kernel $k'$ (standard, **blue**), the invariant kernel $k_G$ (invariant, **green**), and the quasi-invariant kernel $k_G + \varepsilon k'$ (additive, **red**). In all cases, the invariant kernel performs almost as well as the true quasi-invariant kernel.

bustness criteria based on the so-called *safety factor* profile. Using a weighted sum, we reduce the vector objective to $[0, 1]$, with component weights selected to represent the proposed SPR45 design of the STEP tokamak [39, 31]. As the objective components are in direct competition, the highest achievable scores are around $0.7 - 0.8$, while the lowest-scoring converged solutions achieve around $0.4 - 0.5$.

In previous work, the actuator output was represented by arbitrary 1D functions (piecewise linear [31] and Bézier curves [10]). However, in practice the actuator will have a finite number of 'launchers' that drive current in a localised region, and so will be unable to accurately recreate arbitrary profiles. In contrast to previous work, we directly optimise a parameterisation that reflects the actuator's real capabilities: a sum of 12 Gaussians, where each models a launcher targeted at a different point in the plasma cross-section (Figure 5a). In this setting, the objective function is $f : \mathbb{R}^{12} \to \mathbb{R}$, where the input is the normalised radial coordinate of the 12 launchers. As all of the launchers have identical behaviour, the objective is invariant to permutations. However, the full invariant kernel is too costly to compute ($|G| = 12! = 479 \times 10^6$), so we instead use a partially specified kernel (3-block invariant, $|G| = 4! = 24$).

An additional challenge of this task is that large regions of parameter space are *infeasible*, corresponding to simulations that will not converge to a steady-state solution and will therefore be

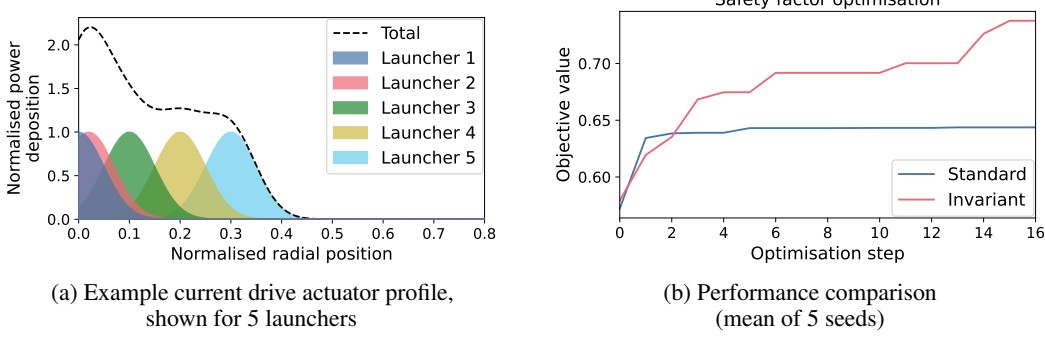

(a) Example current drive actuator profile, shown for 5 launchers

(b) Performance comparison (mean of 5 seeds)

Figure 5: Nuclear fusion application: optimising safety factor by adjusting current drive actuator. In (a), observe that the order of launchers can be permuted without changing the total profile. Incorporating this invariance into the kernel of the UCB algorithm achieves improved performance (b).

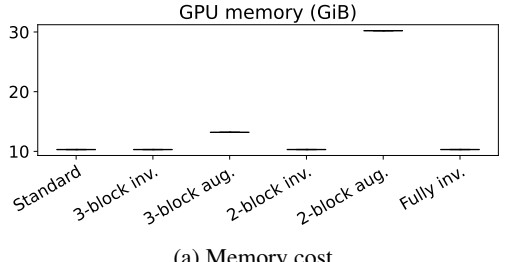
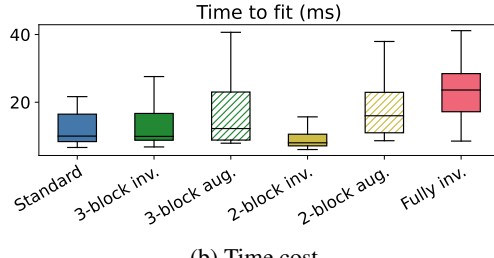

|              |              |
|:------------:|:------------:|
| (a) Memory cost | (b) Time cost |

Figure 6: Effect of group size on cost of data augmentation and invariant kernel methods. Benchmark task is to fit a 6D GP with the given kernel to 100 random datapoints from `PermInv-6D`. Shown are results from 100 seeds, 64 repeats per seed, performed on one NVIDIA V100-32GB GPU using BoTorch. Invariant kernels are (a) more memory efficient than data augmentation, and (b) can be computed faster. Incorporating full invariance via data augmentation exceeds the GPU memory.

impossible to observe. Selecting an appropriate method of dealing with this situation can impact the optimiser's performance, and is an open area of research. In this work, we choose to assign a fixed low score to failed observations for simplicity. As MVR is purely exploratory it has no mechanism that discourages querying points from infeasible regions, leading to many failed observations. UCB is a more natural choice in this setting, as the $\beta$ parameter introduces a balance between querying unknown regions (that could be infeasible) and exploiting known high-scoring regions.

In Figure 5b, we see that the invariant version of UCB achieves significantly improved performance compared to the non-invariant version. In fact, the non-invariant algorithm totally fails to find the small region of high-performance solutions, and instead queries many suboptimal profiles that are similar under the action of permutations. The high sample efficiency of the invariant kernel method will enable tokamak designers to iterate on designs faster, with fewer calls to expensive simulations.

## 5 Limitations

Although the number of terms involved in computing the totally invariant kernel only scales linearly with the group size (Equation (5)), this cost can still become a bottleneck for very large groups; for example, the size of the permutation group scales with the factorial of the dimension, leading to $|G| = 24$ for $d = 4$, but $|G| = 479 \times 10^6$ for $d = 12$. In Figure 6 we compare the compute and memory costs of performing a marginal log likelihood fit of the GP hyperparameters for each of the kernels in the previous section, given 100 observations from the `PermInv-6D` function. The standard, 3-block invariant, and 2-block invariant kernel are faster than the fully invariant kernel by a factor of 2. As we observed that the 2- and 3-block invariant kernels still achieve significantly improved sample efficiency over the standard kernel (Figure 3), we propose that subgroup approximations of the full invariance group can be used as a low-cost alternative when the group size is large, reducing computational cost while maintaining improved performance. Finally, we note that our invariant kernel method scales significantly more favourably than data augmentation, both in terms of memory and compute.

## 6 Conclusion

We have developed new upper bounds on the sample complexity in Bayesian optimization with invariant kernels, highlighting the advantages gained from symmetries that hold for a broad range of practical groups. We also derived novel lower bounds by constructing and quantifying members of the RKHS of invariant functions, extending this approach to the hyperspherical domain and providing a framework for other Riemannian manifolds. Empirically, our experiments show that even partial invariance can effectively improve performance, which is crucial when full invariance is unavailable or computationally intensive. Additionally, we applied these methods to a real-world problem in nuclear fusion engineering, achieving superior outcomes with fewer samples compared to standard methods.

**Broader Impact.** The incorporation of symmetries in Bayesian optimization can have a profound impact on science and engineering. By exploiting symmetrical properties in various applications, such as nuclear engineering, material and drug design, and robotic control, it can reduce the number of samples needed to achieve (near) optimal solutions, thus accelerating research and development processes.

## Acknowledgments

TB was supported by the Engineering and Physical Sciences Research Council (EPSRC) grants EP/T517793/1 and EP/W524335/1. IB was supported by the EPSRC New Investigator Award EP/X03917X/1; EPSRC grant EP/S021566/1; and Google Research Scholar award.

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

# A Proofs

## A.1 Proof of Proposition 1 (RKHS of invariant functions)

The kernel $k$ defines a reproducing kernel Hilbert space, $\mathcal{H}_k$, with inner product $\langle \cdot, \cdot \rangle_{\mathcal{H}_k}$ and norm $\|f\|_{\mathcal{H}_k} = \sqrt{\langle f, f \rangle_{\mathcal{H}_k}}$ for $f \in \mathcal{H}_k$.

Recall from Equation (4) that $S_G(f) = \frac{1}{|G|} \sum_{\sigma \in G} f \circ \sigma$.

We begin by showing that $S_G$ is continuous and bounded. Observe that simultaneous invariance determines the following property of the representer elements $k_x(\cdot) = k(x, \cdot) = k(\cdot, x)$:

$$k_x \circ \sigma = k_{\sigma^{-1}(x)} \quad \forall \sigma \in G. \tag{17}$$

Now observe that:

$$||k_x \circ \sigma||^2 = \langle k_x \circ \sigma, k_x \circ \sigma \rangle = \langle k_{\sigma^{-1}(x)}, k_{\sigma^{-1}(x)} \rangle = k(\sigma^{-1}(x), \sigma^{-1}(x)) = k(x,x) = ||k_x||^2 \tag{18}$$

If $f = \sum_{i=1}^n k_{x_i}$, we have, similarly:

$$||f \circ \sigma||^2 = ||\sum_{i=1}^n k_{\sigma^{-1}(x_i)}||^2 = \sum_{i=1}^n \sum_{j=1}^n k(\sigma^{-1}(x_i), \sigma^{-1}(x_j)) = \sum_{i=1}^n \sum_{j=1}^n k(x_i, x_j) = ||f||^2 \tag{19}$$

Now, let $f_n \to f$ be a Cauchy sequence, where each $f_n$ lies in the span of the $k_x$ elements. We have

$$||f \circ \sigma|| = ||\lim f_n \circ \sigma|| = \lim ||f_n \circ \sigma|| = \lim ||f_n|| = ||f|| \tag{20}$$

Therefore,

$$||S_G f|| \leq \frac{1}{|G|} \sum_{\sigma \in G} ||f \circ \sigma|| \leq ||f|| \tag{21}$$

so $S_G$ is continuous, with $||S_G|| \leq 1$.

We now verify that the image of $S_G$, denoted as $\mathrm{Im}(S_G)$, contains precisely the $G$-invariant functions (Equation (1)). For $f \in \mathcal{H}_k, \sigma \in G$, we have

$$S_G(f) \circ \sigma = \frac{1}{|G|} \sum_{\tau \in G} (f \circ \tau) \circ \sigma \tag{22}$$

$$= \frac{1}{|G|} \sum_{\tau \in G} f \circ (\tau \circ \sigma) \tag{23}$$

$$= \frac{1}{|G|} \sum_{\sigma' \in G} f \circ \sigma' \tag{24}$$

$$= S_G(f), \tag{25}$$

where Equation (23) follows by associativity of $G$, and Equation (24) follows by letting $\sigma' := \tau \circ \sigma$ and using the fact that $G$ is closed under the $\circ$ operator. Hence, $S_G(f)$ is invariant to $\sigma \in G$ (Equation (1)).

Next, we show that the image of $S_G$ is a reproducing kernel Hilbert space with kernel $k_G$ (Equation (5)). First note that $S_G$ is a projection, as applying $S_G$ twice gives

$$S_G(S_G(f)) = \frac{1}{|G|} \sum_{\sigma \in G} S_G(f) \circ \sigma \tag{26}$$

$$= \frac{1}{|G|} \sum_{\sigma \in G} S_G(f) \tag{27}$$

$$= S_G(f), \tag{28}$$

where Equation (27) follows by applying Equation (25).

Moreover, $S_G$ is self-adjoint. Let $f \in \mathcal{H}_k$ be an element in the span of the representer elements $k_x$, such that $f(\cdot) = \sum_{i=1}^{n} \alpha_i k_{x_i}(\cdot)$, and let $g \in \mathcal{H}_k$. Then,

$$\langle S_G(f), g \rangle_{\mathcal{H}_k} = \frac{1}{|G|} \left\langle \sum_{\sigma \in G} f \circ \sigma, g \right\rangle_{\mathcal{H}_k} \tag{29}$$

$$= \frac{1}{|G|} \sum_{i=1}^{n} \sum_{\sigma \in G} \langle \alpha_i k_{\sigma^{-1}(x_i)}, g \rangle_{\mathcal{H}_k} \tag{30}$$

$$= \frac{1}{|G|} \sum_{i=1}^{n} \sum_{\sigma \in G} \alpha_i g(\sigma^{-1}(x_i)) \tag{31}$$

$$= \frac{1}{|G|} \sum_{i} \sum_{\sigma} \langle \alpha_i k_{x_i}, g \circ \sigma^{-1} \rangle_{\mathcal{H}_k} \tag{32}$$

$$= \frac{1}{|G|} \sum_{\sigma} \langle f, g \circ \sigma \rangle_{\mathcal{H}_k} \tag{33}$$

$$= \langle f, S_G(g) \rangle_{\mathcal{H}_k}, \tag{34}$$

where Equation (30) follows by expanding $f$ and recalling that for simultaneously invariant kernels $k_x \circ \sigma = k(\sigma^{-1}(x), \cdot)$, Equation (31) follows by using the reproducing property of $k_{\sigma(x)}$, Equation (32) uses the decomposition of $g \circ \sigma$ in terms of the reproducing element $k_{x_i}$, and Equation (33) uses the definition of $f$.

Now, as before, let $f_n \to f$ be a Cauchy sequence, where $f_n$ lies in the span of the $k_x$ elements. We have

$$\langle \lim f_n, S_G(g) \rangle_{\mathcal{H}_k} = \lim \langle f_n, S_G(g) \rangle = \lim \langle S_G(f_n), g \rangle = \langle \lim S_G(f_n), g \rangle = \langle S_G(\lim f_n), g \rangle_{\mathcal{H}_k}, \tag{35}$$

by continuity of the inner product. Hence, $S_G$ is self-adjoint for all $f, g \in \mathcal{H}_k$.

As $S_G$ is a projection (Equation (28)) and is self-adjoint $\text{Im}(S_G)$ is a closed subspace of $\mathcal{H}_k$. Now, denoting the evaluation functional $L_x : \mathcal{H}_k \to \mathbb{R}$, $L_x(f) = f(x)$, note that since $\text{Im}(S_G)$ is closed, then if $L_x$ was bounded on $\mathcal{H}_k$, it remains bounded on $\text{Im}(S_G)$. Hence $\text{Im}(S_G)$ is itself a RKHS, which we'll denote as $\mathcal{H}'$.

We will now relate $k_{\mathcal{H}'}$, the kernel associated with $\mathcal{H}'$, to $k$ and $G$. Note that, for $f \in \mathcal{H}'$,

$$f(x) = \langle f, k_x \rangle = \langle S_G(f), k_x \rangle = \langle f, S_G(k_x) \rangle. \tag{36}$$

So $S_G(k_x)$ is the reproducing element for $\mathcal{H}'$, giving us

$$k_{\mathcal{H}'}(x, y) = \langle S_G(k_y), S_G(k_x) \rangle = \langle S_G(k_y), k_x \rangle = \frac{1}{|G|} \sum_{\sigma \in G} k(\sigma(x), y), \tag{37}$$

which recovers the formula for the kernel in Equation 5.

Finally, note that $\langle S_G(k_y), S_G(k_x) \rangle$ recovers the more general "double sum" formula for producing a totally invariant kernel out of a kernel which is not necessarily simultaneously invariant [18, 33]:

$$k_G(x, y) = \frac{1}{|G|^2} \sum_{\sigma, \tau \in G} k(\sigma(x), \tau(y)). \tag{38}$$

## A.2 Proof of Theorem 1: upper bound on sample complexity

In this section we derive Theorem 1, the upper bound on regret for Bayesian optimisation algorithms that incorporate invariance. The main elements of this proof were introduced in [41]. Our derivation is similar to the one in section C.1 of [22], but extends that work to more general groups and kernels.

In Appendix A.2.1, we present a concise summary of already known bounds on the information gain of dot-product kernels with polynomial eigendecay on the hypersphere. In Appendix A.2.2, we will adapt this argument for information gain of *totally invariant* versions of these kernels. We then use the bounds on information gain to bound the regret in Appendix A.2.3.

We begin with the following remark which we use throughout this section. Namely, with the assumptions of Theorem 1, a kernel $k$ on $\mathbb{S}^{d-1}$ which is simultaneously invariant with respect to the whole of $O(d)$ is a dot-product kernel. To see this, choose $r \in [-1, 1]$ and pick $x_r, y_r$ such that $\langle x_r, y_r \rangle = r$ and define $\kappa(r) = k(x_r, y_r)$. Observe that $\kappa$ is well defined, as for any other choice of $(x'_r, y'_r)$ there is a (non-unique) isometry $M^{(x'_r, y'_r)}_{(x_r, y_r)}$ such that

$$x'_r = M^{(x'_r, y'_r)}_{(x_r, y_r)} x_r \tag{39}$$

$$y'_r = M^{(x'_r, y'_r)}_{(x_r, y_r)} y_r \tag{40}$$

$$\tag{41}$$

and therefore $k(x_r, y_r) = k(x'_r, y'_r)$. Hence, throughout this section we can consider dot-product kernels only.

### A.2.1 Information gain for kernels on the hypersphere

Consider the hyperspherical domain $\mathbb{S}^{d-1} = \{x \in \mathbb{R}^d \text{ s.t. } \|x\| = 1\}$. The Mercer decomposition of a dot-product kernel on the hypersphere is

$$k(x^T x') = \sum_{k=0}^{\infty} \mu_k \sum_{i=1}^{N(d,k)} Y_{i,k}(x) Y_{i,k}(x'), \tag{42}$$

where $Y_{i,p}$ is the $i$-th spherical harmonic polynomial of degree $k$ and $N(d, k)$ is the number of spherical harmonics of degree $k$ on $\mathbb{S}^{d-1}$, given by

$$N(d, k) = \frac{2k + d - 2}{k} \binom{k + d - 3}{d - 2}, \tag{43}$$

Asymptotically in terms of $k$ we have

$$N(d, k) = \Theta\left(k^{d-2}\right). \tag{44}$$

In this representation, $\mu_k$ can be interpreted as an eigenvalue that is repeated $N(d, k)$ times, with corresponding eigenfunctions $Y_{1,k}, \ldots, Y_{N(d,k),k}$. We let $\psi$ be the maximum absolute value of the eigenfunctions, such that $|Y_{i,k}(x)| \leq \psi \ \forall x \in \mathbb{S}^{d-1}$.

We project onto a subspace of eigenfunctions corresponding to the first $D$ eigenvalues. Let $K$ be the number of distinct values in the first $D$ eigenvalues. It follows that

$$D = \sum_{k=0}^{K} N(d, k) \tag{45}$$

$$c_1 \sum_{k=0}^{K} k^{d-2} \leq D \leq c_2 \sum_{k=0}^{K} k^{d-2} \tag{46}$$

$$C_1 \frac{K^{d-1}}{d-1} \leq D \leq C_2 \frac{K^{d-1}}{d-1} \tag{47}$$

$$D = \Theta\left(K^{d-1}\right) \tag{48}$$

where Equation (46) follows by substituting Equation (44), and Equation (47) follows by bounding the sum with an integral, and $c_i, C_i$ are constants that ensure the asymptotic equalities hold.

Truncating the eigenvalues at $D$ leaves a tail with mass

$$\delta_D \leq \sum_{i=D+1}^{\infty} \lambda_i \psi^2 = \psi^2 \sum_{k=K}^{\infty} \mu_k N(d, k) \tag{49}$$

$$\leq \psi^2 C_1 \sum_{k=K}^{\infty} \mu_k k^{d-2} \tag{50}$$

$$\leq \psi^2 C_2 \sum_{k=K}^{\infty} k^{-\beta_p^*+d-2} \tag{51}$$

$$\leq \psi^2 C_3 K^{-\beta_p^*+d-1} \tag{52}$$

$$\leq \psi^2 C_4 D^{\frac{-\beta_p^*+d-1}{d-1}} \tag{53}$$

where Equation (50) follows by substituting Equation (44), Equation (51) follows if the *distinct* eigenvalues have polynomial decay rate, $\mu_k = \mathcal{O}(k^{-\beta_p^*})$;[2], Equation (52) follows by bounding the sum with an integral; Equation (53) follows because $D = \Theta(K^{d-1})$, and hence $K = \Theta(D^{\frac{1}{d-1}})$.

Theorem 3 in [41] states that

$$\gamma_T \leq \frac{D}{2} \log\left(1 + \frac{\bar{k}T}{\tau D}\right) + \frac{T\delta_D}{2\tau}, \tag{54}$$

where $|k(x, x')| \leq \bar{k} \; \forall x$ and $\tau$ is the observation noise variance. Substituting in Equation (53) gives

$$\gamma_T \leq \frac{D}{2} \log\left(1 + \frac{\bar{k}T}{\tau D}\right) + \frac{T\psi^2 C_4}{2\tau} D^{\frac{-\beta_p^*+d-1}{d-1}} \tag{55}$$

We have freedom in setting $D$. We choose it so that the first term dominates, by setting

$$\frac{D}{2} \log\left(1 + \frac{\bar{k}T}{\tau}\right) \geq \frac{T\psi^2 C_4}{2\tau} D^{\frac{-\beta_p^*+d-1}{d-1}} \tag{56}$$

This is satisfied by

$$D = \left\lceil \left(\frac{T\psi^2 C_4}{\tau \log\left(1 + \frac{\bar{k}T}{\tau}\right)}\right)^{\frac{d-1}{\beta_p^*}} \right\rceil \tag{57}$$

which gives:

$$\gamma_T \leq \left(\left(\frac{T\psi^2 C_4}{\tau \log\left(1 + \frac{\bar{k}T}{\tau}\right)}\right)^{\frac{d-1}{\beta_p^*}} + 1\right) \log\left(1 + \frac{\bar{k}T}{\tau}\right) \tag{58}$$

where we have used the fact that $\lceil x \rceil \leq x + 1$. Hence,

$$\gamma_T = \mathcal{O}\left(T^{\frac{d-1}{\beta_p^*}} \log^{1-\frac{d-1}{\beta_p^*}} T\right). \tag{59}$$

### A.2.2 Information gain for symmetrised kernels on the hypersphere

Our upper bound involves comparing the corresponding reduction in maximal information across the invariant vs the standard kernel. In this section we will use overbars to refer to properties of the symmetrised (invariant) kernel. We remind the reader that $G \leq O(d)$ is a finite subgroup of

---

[2]Matérn kernels on $\mathbb{S}^{d-1}$ satisfy this property with $\beta_p^* = 2\nu + d - 1$ [9].

isometries of the sphere $\mathbb{S}^{d-1}$. Our aim will be to provide bounds on the asymptotics of the eigenspace dimensions $\frac{\bar{N}(d,k)}{N(d,k)}$ where $N(d,k)$ is defined as in A.2.1 and $\bar{N}(d,k)$ is the corresponding eigenspace dimension for the symmetrized kernel $k_G$.

We will follow the exposition from [4], where asymptotic bounds on $\gamma(d,k) := \frac{\bar{N}(d,k)}{N(d,k)}$ are given with two minor differences: 1. our treatment will handle any finite group of isometries and 2. we pay the price for generalizing to arbitrary groups by obtaining less strict decay rates, which will not trouble us for our analysis.

We will show the following:

**Lemma 1.** *With the notations described in this section, assume $G \leq O(d)$ for $d \geq 10$ is a finite group. Then, if $-I \in G$:*

$$0 \leq \gamma(d,k) \leq \frac{C}{k}, \quad \textit{for k odd,} \tag{60}$$

$$\frac{2}{|G|} \leq \gamma(d,k) \leq \frac{2}{|G|} + \frac{C}{k}, \quad \textit{for k even,} \tag{61}$$

$$\tag{62}$$

*and, otherwise:*

$$\frac{1}{|G|} \leq \gamma(d,k) \leq \frac{1}{|G|} + \frac{C}{k} \tag{63}$$

*where $C$ is a constant that depends on $k, d$ and the spectra of elements in $G$.*

*Proof.* Recall from [4], the formula:

$$\gamma(d,k) = \frac{1}{|G|} \sum_{\sigma \in G} \gamma(d,k,\sigma) = \sum_{\sigma \in G} \mathbb{E}_\mu \left[ P_{d,k} \left( \langle \sigma x, x \rangle \right) \right], \tag{64}$$

where $\mu$ is the uniform measure on the sphere $S^{d-1}$, and $P_{d,k}$ is the corresponding Gegenbauer polynomial. To demystify the conditional statement in the Lemma, notice first that $P_{d,k}\left( \langle x, x \rangle \right) = P_{d,k}(1) = 1$, and $P_{d,k}\left( \langle -x, x \rangle \right) = P_{d,k}(-1) = 1$ if $k$ is even and -1 if $k$ is odd. For all other $\sigma \in G$, we'll follow the argument in [4] to show that $\gamma(d,k,\sigma) \to 0$ with asymptotics at least $\mathcal{O}\left( \frac{1}{k} \right)$.

We don't make significant contributions to the method in [4], here, other than to remark on the fact that many of the arguments they present follow through for finite subgroups of $O(d)$.

First remark that for $\sigma \in G \leq O(d)$ there is a matrix representation $A_\sigma$, which admits a canonical basis in which the matrix elements are:

$$\begin{bmatrix} R_1 & & & & & \\ & \ddots & & & 0 & \\ & & R_k & & & \\ & & & \pm 1 & & \\ & 0 & & & \ddots & \\ & & & & & \pm 1 \end{bmatrix} \tag{65}$$

where each $R_\lambda$ is a $2 \times 2$ matrix of the form $\left( \begin{smallmatrix} a & b \\ -b & a \end{smallmatrix} \right)$ with eigenvalues $\lambda$ and $\bar{\lambda}$.

Moreover, notice that since $A_\sigma^{|G|} = I$, all eigenvalues are roots of unity $\lambda = e^{\frac{2\pi i p}{q}}$ with $q \big| |G|$. The the eigenvalues of such matrices are the same as the allowable eigenvalues of permutation matrices considered in [4], from which Proposition 6 gives:

$$\gamma(d,k,\sigma) \leq C k^{-d+s} + o(k^{-d+s}) \tag{66}$$

where $s = \max_{\lambda \in Spec(A_\sigma)} \{ m_\lambda + 4 \cdot \mathbf{1}(|\lambda| < 1) \}$. Now assume $\sigma \neq \pm I$. We have $m_1 \leq d - 2$, $m_{-1} \leq d - 2$ and $m_\lambda \leq \frac{d}{2}$ if $\lambda \neq \pm 1$. Collectively this implies $-d + s \leq -1$ if $d \geq 10$. $\qquad\square$

In the following assume that $-\mathrm{I} \notin G$. The corresponding argument for $-\mathrm{I} \in G$ is similar. In view of Lemma 1

$$\frac{1}{|G|} \leq \frac{\bar{N}(d,k)}{N(d,k)} \leq \frac{1}{|G|} + \frac{C}{k} \tag{67}$$

We take the same number of distinct eigenvalues for the invariant and noninvariant kernels, $K$. However, there will be different multiplicities for the eigenvalues, leading to a different number of eigenvalues $\bar{D}$. The corresponding tail has mass

$$\bar{\delta}_{\bar{D}} \leq \psi^2 \sum_{i=\bar{D}+1}^{\infty} \lambda_i = \psi^2 \sum_{k=K}^{\infty} \mu_k \bar{N}(d,k) \tag{68}$$

$$\leq \psi^2 \sum_{k=K}^{\infty} \mu_k N(d,k) \left( \frac{1}{|G|} + \frac{C}{k} \right) \tag{69}$$

$$\leq \left( \frac{1}{|G|} + \frac{C'}{K} \right) \psi^2 \sum_{k=K}^{\infty} \mu_k N(d,k) \tag{70}$$

$$= \left( \frac{1}{|G|} + \frac{C'}{K} \right) \delta_D \tag{71}$$

$$\leq \left( \frac{1}{|G|} + \frac{C''}{D^{\frac{1}{d-1}}} \right) \delta_D \tag{72}$$

$$\leq \frac{1+\epsilon}{|G|} \delta_D \tag{73}$$

where Equation (69) follows by substituting Equation (67), Equation (70) follows because $\frac{1}{k} < \frac{1}{K}$ for $k > K$, Equation (72) follows because $K = \Theta(D^{\frac{1}{d-1}})$, and for $D > D_\epsilon$ where $D_\epsilon$ satisfies $d! \frac{C''}{D_\epsilon^{\frac{1}{d-1}}} < \epsilon$.

Now, Theorem 3 in [41] gives:

$$\bar{\gamma}_T \leq \frac{\bar{D}}{2} \log \left( 1 + \frac{\bar{k}T}{\tau \bar{D}} \right) + \frac{T\delta_{\bar{D}}}{2\tau}, \tag{74}$$

**Proposition 2.** *If $a_n > 0$, $b_n > 0$, with $\lim a_n = \lim b_n = \infty$, and $\lim b_n/a_n = \lambda > 0$, then $\forall \epsilon > 0$, $\exists N$ such that $\forall N' > N$ we have $\sum_{n<N'} b_n \geq (1-\epsilon)\lambda \sum_{n<N'} a_n$.*

*Proof.* From the definition of the limit, noting that the $1 - \epsilon$ gives slack to bound $\lambda \sum_{n<N} a_n$ above by $\epsilon \sum_{N<n<N'} b_n$. □

Using this proposition on the sequences $N(d,k)$ and $\bar{N}(d,k)$ we have further:

$$\bar{\gamma}_T \leq \frac{D(1+\epsilon)}{2|G|} \log \left( 1 + \frac{\bar{k}T|G|}{\tau D(1+\epsilon)} \right) + \frac{1+\epsilon}{|G|} \frac{T\delta_D}{2\tau} \tag{75}$$

$$\leq \frac{1+\epsilon}{|G|} \left( \frac{D}{2} \log \left( 1 + \frac{\bar{k}T|G|}{\tau D(1+\epsilon)} \right) + \frac{T\delta_D}{2\tau} \right) \tag{76}$$

$$\leq \frac{1+\epsilon}{|G|} \left( \frac{D}{2} \log \left( 1 + \frac{\bar{k}T|G|}{\tau D(1+\epsilon)} \right) + \frac{T}{2\tau} \psi^2 C_4 D^{\frac{-\beta_p^*+d-1}{d-1}} \right) \tag{77}$$

As previously we'll choose $D_T$ as a function of $T$ such that the logarithmic term dominates (notice that since the first term is increasing in $D$ any choice of $D_T$ greater than this will also satisfy), i.e.

$$D_T \log \left( 1 + \frac{\bar{k}T|G|}{\tau(1+\epsilon)} \right) \geq \frac{T}{\tau} \psi^2 C_4 D_T^{\frac{-\beta_p^*+d-1}{d-1}} \tag{78}$$

or

$$D_T \geq \left( \frac{\frac{T}{\tau} \psi^2 C_4}{\log\left(1 + \frac{\bar{k}T|G|}{\tau(1+\epsilon)}\right)} \right)^{\frac{d-1}{\beta_p^*}} \tag{79}$$

With this choice:

$$\bar{\gamma}_T \leq \frac{1+\epsilon}{|G|} \left( \frac{T}{\tau} \psi^2 C_4 \right)^{\frac{d-1}{\beta_p^*}} \log\left( 1 + \frac{\bar{k}T|G|}{\tau(1+\epsilon)} \right)^{\frac{\beta_p^* - d + 1}{\beta_p^*}} \tag{80}$$

Hence,

$$\bar{\gamma}_T = \tilde{\mathcal{O}}\left( \frac{1}{|G|} T^{\frac{d-1}{\beta_p^*}} \right). \tag{81}$$

### A.2.3 From information gain to sample complexity

Under the assumption of sub-Gaussian noise, it is shown in [42, Remark 2] that the maximum variance reduction algorithm presented in Algorithm 1 incurs simple regret satisfying

$$r_T = \mathcal{O}\left( \sqrt{\frac{\gamma_T \log(T^d/\delta)}{T}} \right) \tag{82}$$

with probability at least $1 - \delta$, where $\delta \in (0, 1)$.

Note that we are able to use this result without modification, as our approach only incorporates $G$-invariance through the kernel; hence, any changes to the regret due to the $G$-invariance will only appear in the $\gamma_T$ term. Substituting $\gamma_T$ from Equation (80) into Equation (82) gives

$$r_T = \mathcal{O}\left( \frac{1}{|G|^{\frac{1}{2}}} T^{-\frac{\beta_p^* - d + 1}{2\beta_p^*}} \log^{\frac{\beta_p^* - d + 1}{2\beta_p^*}}(T|G|) \log^{\frac{1}{2}} T/\delta \right) \tag{83}$$

$$= \mathcal{O}\left( \frac{1}{|G|^{\frac{1}{2}}} T^{-\alpha} \log^{\alpha}(T|G|) \log^{\frac{1}{2}} T/\delta \right), \tag{84}$$

where we have absorbed factors that do not depend on $T$ or $G$, and let $\alpha = \frac{\beta_p^* - d + 1}{2\beta_p^*}$.

Our goal is for $r_T = \mathcal{O}(\epsilon)$. This can be achieved by setting

$$T \propto \frac{1}{|G|^{\frac{1}{2\alpha}}} \frac{1}{\epsilon^{\frac{1}{\alpha}}} \log\left( \frac{1}{|G|^{\frac{1}{2\alpha}-1}} \frac{1}{\epsilon^{\frac{1}{\alpha}}} \right) \log^{\frac{1}{2\alpha}}\left( \frac{1}{|G|^{\frac{1}{2\alpha}}} \frac{1}{\epsilon^{\frac{1}{\alpha}}} \right) \tag{85}$$

with appropriate implied constants. Substitution of $T$ from Equation (85) into Equation (84) makes it evident that for this $T$ we have

$$r_T = \mathcal{O}(\epsilon) + \mathcal{O}(\log\log T + \log\log|G|). \tag{86}$$

Substituting $\alpha$ into Equation (85) gives the final upper bound on sample complexity, which is

$$T = \mathcal{O}\left( \left( \frac{1}{|G|} \right)^{\frac{\beta_p^*}{\beta_p^* - d + 1}} \epsilon^{-\frac{2\beta_p^*}{\beta_p^* - d + 1}} \log\left( \frac{\epsilon^{-\frac{2\beta_p^*}{\beta_p^* - d + 1}}}{|G|^{\frac{d-1}{\beta_p^* - d + 1}}} \right) \log^{\frac{\beta_p^*}{\beta_p^* - d + 1}}\left( \frac{\epsilon^{-\frac{2\beta_p^*}{\beta_p^* - d + 1}}}{|G|^{\frac{\beta_p^*}{\beta_p^* - d + 1}}} \right) \right). \tag{87}$$

For the Matérn kernel on the hypersphere, [9] showed that

$$\beta_p^* = 2\nu + d - 1, \tag{88}$$

and hence, the MVR algorithm with an invariant Matern GP defined on $\mathbb{S}^{d-1}$ will achieve simple regret $\epsilon$ with probability $1 - \delta$ if

$$T = \tilde{\mathcal{O}}\left( \left( \frac{1}{|G|} \right)^{\frac{2\nu + d - 1}{2\nu}} \epsilon^{-\frac{2\nu + d - 1}{\nu}} \right), \tag{89}$$

where $\tilde{\mathcal{O}}$ hides logarithmic factors in $\epsilon$ and $|G|$.

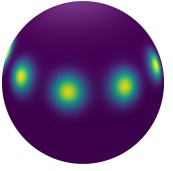
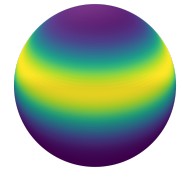

Figure 7: Examples of invariant functions on the sphere. The construction for our lower bound consists of functions invariant to finite rotations, as in Figure 7a, but does not include functions like Figure 7b (which produce different packings).

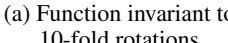

(a) Function invariant to 10-fold rotations

(b) Function invariant to all rotations along an axis

### A.3 Proof of Theorem 2: lower bound on sample complexity

In this section, we derive Theorem 2, the lower bound on regret for the invariant Matérn kernel under permutation groups. The main elements of this proof were introduced in [12], which extended the method in [35]. This constructive method serves to demonstrate the tightness of the upper bound. As in [12] and [42], our method involves building up a set of candidate optimization targets in $\mathcal{H}_{k_G}$. However, unlike these papers, it is no longer clear in this setting that such constructions are optimal in terms of the resulting dependence of the lower bound on $|G|$ (the current gap to the upper bound potentially measures this failure rate). Our methods will produce elements in $S_G(\mathcal{F}) \subseteq \mathcal{H}_{k_G}$, for a particular set of functions $\mathcal{F}$, and it's the object of further research if these constructions can be improved.In Figure 7a, we have an example of the types of functions we construct, by applying the operator $S_G$ to a simple bump function. In this example, $G$ is a group of rotations along the vertical axis. In Figure 7b is an example of a function we cannot retrieve through our method. This function is similarly invariant (also to all other rotations along the axis). Our method relies on a packing argument for the supports of such functions and it is a priori unclear how the tightest packing can be obtained.

We identify a bounded subset of the RKHS of invariant functions that contains functions with the property that each point in $\mathcal{X}$ is $\epsilon$-optimal for at most one function. We then construct a set of noisy bandit problems using functions from this class, before applying [12, Lemma 1] to lower-bound the number of samples that a learner requires in order to distinguish between them, and hence the best-case regret for this configuration.

#### A.3.1 An explicit construction

The proof for our lower bound will be constructive. We will exhibit a family of functions for which there is a computable lower bound of samples needed to distinguish their maxima. The challenge comes from constructing these functions *natively* in the RKHS of invariant functions. The strategy we employ here is to construct functions by symmetrization of simple bump functions. For the benefit of the reader, we'll show an explicit version of this construction first, on the hypercube $\mathcal{X} = [0,1]^d$ equipped with the action of $G \leq S_d$ and subsequently extend to general actions of subgroups of $O(d)$ on the sphere $\mathbb{S}^{d-1}$.

**Lemma 2** (Bounded support function construction [12, Lemma 4]). *Let* $h(x) = \exp\left(\frac{-1}{1-\|x\|^2}\right) \mathbb{1}\{\|x\|_2 < 1\}$ *be the d-dimensional bump function, and define* $g(x, w, \epsilon) = \frac{2\epsilon}{h(0)} h(\frac{x}{w})$ *for some* $w > 0$ *and* $\epsilon > 0$. *Then, g satisfies the following properties:*

1. $g(x, w, \epsilon) = 0$ *for all x outside the* $\ell_2$-*ball of radius w centred at the origin;*

2. $g(x, w, \epsilon) \in [0, 2\epsilon]$ *for all x, and* $g(0) = 2\epsilon$;

3. $\|g\|_k \leq c_1 \frac{2\epsilon}{h(0)} \left(\frac{1}{w}\right)^{\nu} \|h\|_k$ *when k is the Matérn-$\nu$ kernel on* $\mathbb{R}^d$, *where* $c_1$ *is constant. In particular, we have* $\|g\|_k \leq B$ *when* $w = \left(\frac{2\epsilon c_1 \|h\|_k}{h(0)B}\right)^{\frac{1}{\nu}}$.

Let $e_1, \ldots, e_d$ be the canonical basis vectors in $\mathbb{R}^d$. Then, the set

$$L(\alpha) = \left\{ \frac{2\lambda_1 - 1}{2} \alpha e_1 + \cdots + \frac{2\lambda_d - 1}{2} \alpha e_d \; : \; \lambda_1, \ldots, \lambda_d \in \mathbb{Z} \right\} \tag{90}$$

forms a lattice of points on $\mathbb{R}^d$ with spacing $\alpha$. Equivalently, $L(\alpha)$ defines a partitioning of $\mathbb{R}^d$ into disjoint regions or 'lattice cells' $\{\mathcal{R}(x_0) : x_0 \in L(\alpha)\}$, where $\mathcal{R}(x_0)$ is a $d$-dimensional hypercube of side length $\alpha$ centred on $x_0$.

Define the function class $\mathcal{F}(w, \epsilon) = \{g(x - x_0, w, \epsilon) : x_0 \in L(w)\}$. By property (1) of Lemma 2, $g(x - x_0, w, \epsilon)$ is zero outside of $\mathcal{R}(x_0)$. Hence, the support of functions $f \in \mathcal{F}(w, \epsilon)$ are pairwise disjoint. In addition, as $\mathcal{F}(w, \epsilon)$ consists only of translates of $g(x, w, \epsilon)$, we retain $\max f = 2\epsilon$ and $\|f\|_k \leq B \; \forall f \in \mathcal{F}(w, \epsilon)$ for appropriate $w$ from properties 2 and 3 of Lemma 2 respectively.

The hypercube $[0, 1]^d$ contains $N$ of these functions, where

$$N = \left\lfloor \frac{1}{w} \right\rfloor^d = \left\lfloor \frac{h(0)B}{2\epsilon c_1 \|h\|_k} \right\rfloor^{d/\nu}. \tag{91}$$

Now, we construct a set of *G-invariant* bump functions with disjoint support on the hypercube $[0, 1]^d$, centred on the lattice cells for a class of groups $G$ equipped with an action which we define below.

**Lemma 3** (Set of invariant functions). *Let $G$ be a subset of $S_d$, the permutation group on $d$ elements. $G$ acts on $\mathbb{R}^d$ (and on any invariant subset $\mathcal{X}$) by permutation of the coordinate axes. Let $\mathcal{F}(w, \epsilon) = \{g(x - x_0, w, \epsilon) : x_0 \in L(w)\}$. Then, $\bar{\mathcal{F}}(w, \epsilon, G) = \{S_G(f) : f \in \mathcal{F}(|G|^{1/\nu}w, |G|\epsilon)\}$, where $S_G$ is defined as in Proposition 1, satisfies the following properties:*

1. *$\bar{\mathcal{F}}(w, \epsilon, G)$ is a set of $G$-invariant functions on $\mathbb{R}^d$ with pairwise disjoint support;*

2. *$\max_x \bar{f}_i(x) = \frac{2|G|\epsilon}{|G \cdot x_i|}$, where $\bar{f}_i(x) = S_G(g(x - x_i, |G|^{1/\nu}w, |G|\epsilon)) \in \bar{\mathcal{F}}(w, \epsilon, G)$;*

3. *$\|\bar{f}\| \leq B \quad \forall \bar{f} \in \bar{\mathcal{F}}(w, \epsilon, G)$.*

*Proof.* First, we show that $\bar{\mathcal{F}}(w, \epsilon, G)$ is indeed a $S_G$-invariant set of $G$-invariant functions with disjoint support. As all translates $g(x - x_0)$ are symmetric around $x_0$, it holds that $g(\sigma(x) - x_0) = g(x - \sigma^{-1}(x_0))$. From Equation (90), it is evident that the lattice $L(\alpha)$ is invariant to permutations; that is, $\sigma(x_0) \in L(w)$ for all $x_0 \in L(w), \sigma \in G$. Hence, $\mathcal{F}(w, \epsilon)$ itself is invariant under $S_G$, and a fortiori $\bar{\mathcal{F}}(w, \epsilon, G) = S_G\left(\mathcal{F}(|G|^{1/\nu}w, |G|\epsilon)\right)$.

Next, we show that functions in $\bar{\mathcal{F}}(w, \epsilon, G)$ have maximum value $\frac{2|G|\epsilon}{|G \cdot x_i|}$. Let $f_i \in \mathcal{F}(|G|^{1/\nu}w, |G|\epsilon)$ be the function centred on $x_i$. From Lemma 2, property 2, $f_i$ has maximum value $2|G|\epsilon$. Let $G_{x_i} = \{\sigma : \sigma(x_i) = x_i\}$ denote the stabilizer group of $G$ at $x_i$.

Since the lattice itself is invariant, the image $G \cdot x_i$ of the centre will have size

$$|G \cdot x_i| = \frac{|G|}{|G_{x_i}|}. \tag{92}$$

The preimage of any element in $G \cdot x_i$ consists of exactly $|G_{x_i}|$ elements. Therefore,

$$S_G(f_i) = \frac{1}{|G|} \sum_{H \in G/G_{x_i}} \sum_{\sigma \in H} f_i \circ \sigma \tag{93}$$

$$= \frac{|G_{x_i}|}{|G|} \sum_{H \in G/G_{x_i}} f_i \circ \sigma_H, \tag{94}$$

$$\tag{95}$$

where $\sigma_H$ is a representative of the coset $H$ and $f_i \circ \sigma_H$ is the unique value of $f_i \circ \sigma$ on this coset, regardless of its representative.

Note that for any cosets $H$ and $K$, $f_i \circ \sigma_H$ and $f_i \circ \sigma_K$ have disjoint supports from the same arguments as before, and thus,

$$\max S_G(f_i) = \frac{|G_{x_i}|}{|G|} \max f_i \circ \sigma = 2|G|\epsilon \frac{|G_{x_i}|}{|G|} = \frac{2|G|\epsilon}{|G \cdot x_i|}. \tag{96}$$

Finally, we show that functions in $\bar{\mathcal{F}}(w, \epsilon, G)$ have norm bounded by $B$. Let $\bar{f} \in \bar{\mathcal{F}}(w, \epsilon, G)$ and $f \in \mathcal{F}(|G|^{1/\nu}w, |G|\epsilon)$. From property (3) in Lemma 2,

$$\|f\|_k \leq \frac{c_1 2|G|\epsilon}{h(0)} \left(\frac{1}{|G|^{1/\nu}w}\right)^\nu \|h\|_k$$
$$= \|g\|_k \leq B,$$

where $w$ is the same as before. From Proposition 1, recall that $S_G$ is a projection; hence, $\|f\|_k \leq B$ implies $\|S_G(f)\|_{k_G} \leq B$, and so $\|\bar{f}\|_{k_G} \leq \|f\|_k \leq B$. $\qquad \square$

We now restrict the function set to the unit hypercube. The number of functions from $\mathcal{F}(|G|^{1/\nu}w, |G|\epsilon)$ (the set of rescaled non-invariant functions) that fit in the unit hypercube is given by $N'$, where

$$N' = \left\lfloor \frac{1}{|G|^{1/\nu}w} \right\rfloor^d \tag{97}$$

$$= \left\lfloor \frac{1}{|G|^{1/\nu}} \right\rfloor^d N, \tag{98}$$

where $N$ is defined in Equation (91).

Clearly, the number of orbits in $\mathcal{F}(|G|^{1/\nu}w, |G|\epsilon)$, i.e. the number of elements in $\bar{\mathcal{F}}(w, \epsilon, G)$ is equal to the number of orbits of $G$ on $L(w)$, by Lemma 3. We would like to count the number of orbits, or at least bound it from below. For specific groups $G$ one can do this explicitly, however, in general it is impossible to give a count of the orbits, without further assumption. What we can, however say is that the number of orbits of size exactly $|G|$ represents almost the total number of orbits when $w$ is small.

**Proposition 3.** *Let $M = |\bar{\mathcal{F}}(w, \epsilon, G)| = \left|\left(L(|G|^{1/\nu}w) \bigcap [0,1]^d\right)/G\right|$. Let $K := \lfloor \frac{1}{|G|^{1/\nu}w} \rfloor$. Then $M \geq \frac{K^d}{|G|}$, and this bound is asymptotically tight in the limit of $w \to 0$.*

*Proof.* Let $\mathcal{W}(s)$ be the set of points in $L(|G|^{1/\nu}w) \bigcap [0,1]^d$ with exactly $s \leq d$ repeated indices. Assume $w$ is such that $K := \lfloor \frac{1}{|G|^{1/\nu}w} \rfloor > d + 1$. Then, $\mathcal{W}(0) \neq \emptyset$ under the action of $G$, such points in $L(w)$ will have $|G \cdot x| = |G|$.

Points in $\bigcup_{s=1}^d \mathcal{W}(s)$, i.e. points with repeated indices lie in a union of manifolds of dimension $d-1$ or below within the hypercube $[0,1]^d$. As such, $w$ decreases,

$$\frac{|\mathcal{W}(0)|}{N'} \to 1. \tag{99}$$

This is easy to see as

$$|\mathcal{W}(0)| = K \cdot (K-1) \cdot ...(K-d). \tag{100}$$

Hence, we have that $|\bar{\mathcal{F}}(w, \epsilon, G)|$ approaches $\frac{|\mathcal{F}(|G|^{1/\nu}w, |G|\epsilon)|}{|G|}$ from above as the width $w$ tends to 0:

$$\lim_{w \to 0} \frac{|\bar{\mathcal{F}}(w, \epsilon, G)|}{|\mathcal{F}(|G|^{1/\nu}w, |G|\epsilon)|} = \frac{1}{|G|}. \tag{101}$$

$\qquad \square$

Note that $M$ is also the number of functions from $\bar{\mathcal{F}}(w, \epsilon, G)$ that are supported in $[0,1]^d$. Then, we can rewrite the above inequality as:

$$M \geq \frac{N'}{|G|} \tag{102}$$

$$\geq \frac{1}{|G|^{1+d/\nu}} N, \tag{103}$$

where $N$ is the number of functions from $\mathcal{F}(w, \epsilon)$ that fit in $[0, 1]^d$, as defined in Equation (91). Equation (103) gives a direct relationship between the packing of invariant and non-invariant bump functions with bounded norm that fit in the unit hypercube.

Note that the above analysis shows that this bound is tight for small enough $w$ which depends only on $d$ and $\nu$ since $|G| < d!$. If one is interested only in Equation (103), this is easy to obtain as $M$, the number of orbits is greater than $N'$ divided by the size of the largest orbit.

Note also that in the case of noiseless observations any optimisation algorithm will have to query at least $M$ (resp. $N$) points to be able to distinguish between functions in $\bar{\mathcal{F}}(w, \epsilon, G)$ (resp. $\mathcal{F}(w, \epsilon)$) defined on $[0, 1]^d$; hence, Equation (103) (resp. Equation (91)) can also be interpreted as a lower bound on the sample complexity of finding the maximum of these functions.

### A.3.2 Distinguishing between bandit instances

The subsequent section of the proof mirrors the approach used to derive a simple regret lower bound in the standard (non-invariant) setting, as detailed in [12, Section 4.2]. The primary distinction lies in the construction of the representative functions that exhibit group invariance, as outlined in Equation (104) and Equation (105).

Recall the definition of a region $\mathcal{R}(\cdot)$ from the discussion following Equation (90). Define $\bar{\mathcal{R}}_i$ as $\bar{\mathcal{R}}_i = \bigcup_{\sigma \in G} \mathcal{R}(\sigma(x_i))$, where $x_i \in L(|G|^{1/\nu} w)$ is the center of the region. Note that these regions are self-contained under $G$, so that $x \in \bar{\mathcal{R}}_i \implies \sigma(x) \in \bar{\mathcal{R}}_i$, and are pairwise disjoint. Therefore, we define a set of $M$ such regions $\{\bar{\mathcal{R}}_i\}_{i=1}^M$, which partition our domain. The value of $M$ is lower bounded in Equation (103).

We replace $B$ with $B/3$ in the construction of $\mathcal{F}(|G|^{1/\nu} w, |G|\epsilon)$ (see Lemma 3).[3] Let $f_i$ and $f_j$ ($i \neq j$) be shifted bump functions from the set $\mathcal{F}(|G|^{1/\nu} w, |G|\epsilon)$, centered at some fixed $x_i$ and $x_j$ respectively, where $x_i, x_j \in L(|G|^{1/\nu} w)$. Furthermore, these are selected such that the sizes of the orbits, $|G \cdot x_i|$ and $|G \cdot x_j|$, are both equal to $|G|$. Then, we construct the pair

$$\bar{f} = S_G(f_i), \tag{104}$$

$$\bar{f}' = S_G(f_i) + 2 S_G(f_j). \tag{105}$$

The following holds $\|\bar{f}\|_{k_G} \leq B/3$ and $\|\bar{f}'\|_{k_G} \leq B$ (from Lemma 3 and triangle inequality). We observe that $\bar{f}$ and $\bar{f}'$ coincide outside $\bar{\mathcal{R}}_j$ i.e., $\bar{f}' - \bar{f}$ is only non-zero in region $\bar{\mathcal{R}}_j$. Moreover, we have $\max_{x \in \bar{\mathcal{R}}_i} \bar{f}(x) = 2\epsilon$ and $\max_{x \in \bar{\mathcal{R}}_j} \bar{f}'(x) = 4\epsilon$. It follows that only points from $\bar{\mathcal{R}}_j$ and $\bar{\mathcal{R}}_i$ can be $\epsilon$-optimal for $\bar{f}'$ and $\bar{f}$, respectively.

In the subsequent steps, let $P_f[\cdot]$ and $\mathbb{E}_f[\cdot]$ denote probabilities and expectations with respect to the random noise, given a suitably chosen underlying function $f$. Additionally, we define $P_f(y|x)$ as the conditional distribution $N(f(x), \sigma^2)$, consistent with the Gaussian noise model.

We fix $T \in \mathbb{Z}^+$, $\delta \in (0, 1/3)$, $B > 0$ and $\epsilon \in (0, 1/2)$, and assume the existence of an algorithm such that for any $f \in \mathcal{F}_{k_G}(B)$, it achieves an average simple regret $r(x^{(T)}) \leq \epsilon$ with a probability of at least $1 - \delta$. Here, $x^{(T)}$ represents the final point reported by the algorithm. Note that an algorithm that achieves simple regret less than $\epsilon$ for all $f \in \mathcal{F}_{k_G}(B)$ automatically solves the decision problem of distinguishing between $f$ and $f'$ as above. If no such algorithm exists, the sample complexity necessary to achieve this bound on regret must be higher than $T$. Next, we establish a lower bound on the number of samples required for such an algorithm to differentiate between two bandit environments (with reward functions $\bar{f}$ and $\bar{f}'$ and the same Gaussian noise model) that share the same input domain but have distinct optimal regions.

Let $A$ denote the event that the returned point $x^{(T)}$ falls within the region $\bar{\mathcal{R}}_i$. Assume that an algorithm achieves a simple regret of at most $\epsilon$ for both functions $\bar{f}$ and $\bar{f}'$ (both $\bar{f}, \bar{f}' \in \mathcal{F}_{k_G}(B)$ as established before) each with a probability of at least $1 - \delta$. Under our assumption on simple regret, this implies $P_{\bar{f}}[A] \geq 1 - \delta$ and $P_{\bar{f}'}[A] \leq \delta$ since only points within $\bar{\mathcal{R}}_i$ can be $\epsilon$-optimal under $\bar{f}$, and only points within $\bar{\mathcal{R}}_j$ can be $\epsilon$-optimal under $\bar{f}'$.

---

[3]We note that this modification affects only the constant factors (also in the bound on $M$ from Equation (103)), which are not important for our analysis.

We are now in position to apply [12, Lemma 1]. We apply this lemma using the previously defined $\bar{f}$, $\bar{f}'$, and $A$.[4] It therefore holds that, for $\delta \in (0, \frac{1}{3})$:

$$\sum_{m=1}^{M} \mathbb{E}_{\bar{f}}[C_m] \max_{x \in \bar{\mathcal{R}}_m} \text{KL}\left[\mathbb{P}_{\bar{f}}(\cdot|x) \,||\, \mathbb{P}_{\bar{f}'}(\cdot|x)\right] \geq \log \frac{1}{2.4\delta}, \tag{106}$$

where $C_m$ is the number of queried points in $\bar{\mathcal{R}}_m$ up to time $T$.

For the two bandit instances and any input $x$, the observation distributions are $\mathcal{N}(\bar{f}(x), \sigma^2)$ and $\mathcal{N}(\bar{f}'(x), \sigma^2)$. Utilising the standard result for the KL divergence between two Gaussian distributions with the same variance, we obtain:

$$\max_{x \in \bar{\mathcal{R}}_m} \text{KL}\left[\mathbb{P}_{\bar{f}}(\cdot|x) \,||\, \mathbb{P}_{\bar{f}'}(\cdot|x)\right] = \frac{(\max_{x \in \bar{\mathcal{R}}_m} \bar{f}(x) - \max_{x \in \bar{\mathcal{R}}_m} \bar{f}'(x))^2}{2\sigma^2} \tag{107}$$

$$= \begin{cases} \frac{8\epsilon^2}{\sigma^2} & m = j \\ 0 & \text{otherwise,} \end{cases} \tag{108}$$

as $\bar{f}(x) = \bar{f}'(x)$ for $x \notin \bar{\mathcal{R}}_j$, and for $x \in \bar{\mathcal{R}}_j$ we have that $\bar{f}(x) = 0$ and $\max_{x \in \bar{\mathcal{R}}_j} \bar{f}'(x) = 4\epsilon$.

Hence, Equation (106) becomes

$$\frac{8\epsilon^2}{\sigma^2} \mathbb{E}_{\bar{f}}[C_j] \geq \log \frac{1}{2.4\delta}. \tag{109}$$

Let $J = \{j \in I | x_j \in \mathcal{W}(0)\}$ be the set of indices $j$ such that $f_j$ is centred around a point with no repeated indices such as in Proposition 3. Let $\rho_w = \frac{|J|}{|G|M}$ be the fraction of all orbits of size exactly $|G|$. By Proposition 3, for any small $\delta_w$, there is a correspondingly small enough $w$ such that $\rho_w \geq 1 - \delta_w$. Since the previous inequality holds for an arbitrary $j$, we sum over over $j \in (J \setminus G\{i\})/G$ to arrive at:

$$\frac{1}{\frac{|J|}{|G|} - 1} \sum_{j \neq i} \frac{8\epsilon^2}{\sigma^2} \mathbb{E}_{\bar{f}}[C_j] \geq \log \frac{1}{2.4\delta}. \tag{110}$$

By rearranging and noting that $\sum_{j=1}^{M} \mathbb{E}_{\bar{f}}[C_j] = T$ we obtain

$$T \geq (M\rho_w - 1)\frac{\sigma^2}{8\epsilon^2} \log \frac{1}{2.4\delta}. \tag{111}$$

Substituting in $M$ from Equation (103) gives

$$T \geq \frac{1 - \delta_w}{|G|^{\frac{\nu+d}{\nu}}} N \frac{\sigma^2}{8\epsilon^2} \log \frac{1}{2.4\delta} - \frac{\sigma^2}{8\epsilon^2} \log \frac{1}{2.4\delta}, \tag{112}$$

and using $N$ from Equation (91) gives the final bound

$$T = \Omega\left(\frac{1}{|G|^{\frac{\nu+d}{\nu}}} \frac{\sigma^2}{\epsilon^2} \left(\frac{B}{\epsilon}\right)^{d/\nu} \log \frac{1}{\delta}\right), \tag{113}$$

where the implied constants can depend on $d$, $l$, and $\nu$.

## A.4  Lower bounds on different underlying spaces.

In this section we'll investigate how to raise the lower bounds on sample complexity to embedded submanifolds $\mathcal{X} \hookrightarrow \mathbb{R}^d$. The full treatment below is valid only for the hypersphere, $\mathcal{X} = \mathbb{S}^d$ but many of the ingredients required are valid on other manifolds as well. When specific statements about the sphere are made in the treatment below, we will explicitly refer to $\mathbb{S}^d$ instead of $\mathcal{X}$.

---

[4]Following the approach in [12], we deterministically set the stopping time in Lemma 1 to $T$, corresponding to the fixed-length setting in which the time horizon is pre-specified. For an adaptation to scenarios where the algorithm may determine its stopping time, refer to Remark 1 in [12].

In the proof in A.3, we used the existence of a $G$-invariant lattice to underpin our bump function construction. Moreover, we've used the fact that the RKHS norm $B := \|g\|_k \sim \frac{\epsilon}{w^\nu}$. We will replicate the underlying function class construction for $\bar{\mathcal{F}}(w, \epsilon, G)$ here.

We will recover the same property for bump functions on $\mathcal{X}$ but under a different notion of rescaling.

Pick a point $p \in U \subset \mathcal{X}$ and consider $(y^1, ..., y^m)$ a system of coordinates on $U$ so that $x_i(y^1, ..., y^m)$ represent the standard coordinates on the ambient space $\mathbb{R}^d$. Let $f_{p,w,\epsilon} : \mathbb{R}^d \to \mathbb{R}_+$ be a bump function around $p$ which is 0 outside $B_w(p; \mathbb{R}^d)$ and supported in $B_{w/2}(p; \mathbb{R}^d)$ with height $\epsilon$, as in section A.3. Let $\phi_{p,w,\epsilon} = f_{p,w,\epsilon}\big|_{\mathcal{X}}$.

If $\mathcal{X} = \mathbb{S}^d$, then the intersection $B_{r_w}(p; \mathcal{X}) := B_w(p; \mathbb{R}^d) \bigcap \mathcal{X}$ is a geodesic neighbourhood with geodesic radius $r_w = \arccos\left(\frac{2-w^2}{2}\right)$. For small enough $w$,

$$w \le r_w \le w + w^2, \tag{114}$$

by computing the Taylor expansion.

We will lso need to compute the RKHS norm for $\phi_{p,w,\epsilon}$ for some suitably defined kernel. Let $k_\nu : \mathbb{R}^d \times \mathbb{R}^d$ be the standard Matérn kernel, and consider $\tilde{k}_\nu$ its restriction to $\mathcal{X}$.

Next, we recall the existence of a fundamental domain for the action of finite groups $G$ on $\mathcal{X}$. Following the definition of fundamental domain from [21], we say $F \subset \mathcal{X}$ is a fundamental domain for the action of $G$ if:

- $F$ is a domain.
- $G \cdot \bar{F} = \mathcal{X}$
- $gF \bigcap F \ne \emptyset \implies g = e$, the identity.
- For any compact $K \subset \mathcal{X}$, the *transporter set* $(\bar{F}|K)_G$ is finite.

Note that the last condition is trivial if $G$ is finite. The construction of $F$ is simple given that $G$ acts by isometries of the underlying space $\mathbb{R}^d$ and $g_\mathcal{X} = \iota^* g_{\mathbb{R}^d}$.

Indeed, let $x$ be a point such that $|Gx| = |G|$ (such a point trivially exists, as each member of $G$ other than the identity fixes a subspace of dimension at most $d - 1$) on the $\mathcal{X} = \mathbb{S}^{d-1}$. Then, $|Gx| = |G|$, and the Dirichlet domain:

$$F_x = \{y \in \mathcal{X} | d_\mathcal{X}(x, y) < d_\mathcal{X}(gx, y), \quad \forall g \in G \setminus \{e\}\} \tag{115}$$

is a fundamental domain for the action (see [21] for proof). One can see this set of points as the interior of the points in $\mathcal{X}$ which are closest to $x$ rather than any of its other translates under $G$. For example, for the action of $S_d$ on the positive orthant $\mathbb{S}^{d-1}$, it is any of the simplices in the barycentric subdivision.

Equipped with this definition, we construct the lattice of bump function centres on $\mathcal{X}$ as follows. First construct a $2r_w$-net $L(r_w; F_x)$ on $F_x$ as the set of centres of an optimal geodesic ball packing of $F_x$. Then define

$$L(r_w) := L(r_w; \mathcal{X}) = G \cdot L(r_w; f_x) \tag{116}$$

We will need to compare the cardinalities of $L(r_w)$ and $L(tr_w)$ for some scaling factor $t > 0$. Let $\Pi(r_w)$ be the packing number of $F \subseteq \mathbb{S}^{d-1}$ by geodesic balls of radius $r_w$.

We have trivially that

$$\frac{Vol(F)}{Vol(B_{2r_w}(p; \mathcal{X}))} \le \Pi(r_w) \le \frac{Vol(F)}{Vol(B_{r_w}(p; \mathcal{X}))} \tag{117}$$

where the first inequality comes trivially from noticing that a maximal $r_w$ packing is a $2r_w$ covering. Moreover, since the exponential map is a diffeomorphism on small enough neighbourhoods (with derivative I at the origin), we have that:

$$K_1 Vol(B_{r_w}(p; \mathbb{R}^{d-1})) \le Vol(B_{r_w}(p; \mathcal{X})) \le K_2 Vol(B_{r_w}(p; \mathbb{R}^{d-1})) \tag{118}$$

for some $0 < K_1 < K_2$. Combining equations 117 and 118 gives:

$$c_1 t^{d-1} \Pi(r_w) \leq \Pi(t r_w) \leq c_2 t^{d-1} \Pi(r_w) \tag{119}$$

Following this discussion, we have the following lemma which is an extension to Lemma 3. Denote by $w_r$ the inverse mapping of $w \to r_w$, defined on small enough geodesic balls on $\mathbb{S}^{d-1}$. We write $\mathcal{F}(w, \epsilon) = \{f_{p,w,\epsilon} : p \in L(r_w)\}$ for a set of functions on the ambient space and $\mathcal{F}_{\mathcal{X}}(r_w, \epsilon) = \{\phi_{p,w,\epsilon} := f_{p,w,\epsilon}\big|_{\mathcal{X}} : p \in L(r_w)\}$ for the restriction $\mathcal{F}(w, \epsilon)\big|_{\mathcal{X}}$. Similarly, we write $\bar{\mathcal{F}}(w, \epsilon, G) = \{S_G(f) : f \in \mathcal{F}(|G|^{1/\nu} w, |G| \epsilon)\}$ and $\bar{\mathcal{F}}_{\mathcal{X}}(w, \epsilon, G) = \bar{\mathcal{F}}(w, \epsilon, G)\big|_{\mathcal{X}}$

**Lemma 4** (Invariant functions on the sphere). *The sets* $\bar{\mathcal{F}}_{\mathcal{X}}(r_w, \epsilon, G)$ *and* $\bar{\mathcal{F}}(w, \epsilon, G)$ *satisfy the following properties.*

1. $\bar{\mathcal{F}}(w, \epsilon, G)$ *is a set of $G$-invariant functions on $\mathbb{S}^{d-1}$ with pairwise disjoint support;*

2. $\max_x S_G(f_{p,w,\epsilon})(x) = \frac{2|G|\epsilon}{|G \cdot p|}$.

3. $\|\bar{f}\| \leq B \quad \forall \bar{f} \in \bar{\mathcal{F}}_{\mathcal{X}}(w, \epsilon, G)$.

*Proof.* The key ingredient for 1. is to note that each $f_{p,w,\epsilon}(x) = g(\frac{x-p}{w})$, and thus, each $f$ is $G$-invariant around the point $p$, namely, for $\sigma \in G$, $f_{p,w,\epsilon} \circ \sigma = f_{\sigma^{-1}(p),w,\epsilon}$. Then, since $L(r_w)$ is $G$-invariant by construction, 1. follows. For 2. the proof is identical to Lemma 3 2. For 3. notice that

$$\|\phi_{p,tw,\epsilon}\|_{\tilde{k}_\nu} \leq \inf_{\phi^e} \|\phi^e_{p,tw,\epsilon}\|_{k_\nu} \leq \|f_{p,tw,\epsilon}\|_{k_\nu} = t^\nu \|f_{p,w,\epsilon}\|_{k_\nu} \tag{120}$$

where the second inequality follows from the restriction and $\phi^e$ is any extension of $\phi$ to the underlying space, since the norm of a restriction is less than that of all extensions.

$\square$

Let $N = |\mathcal{F}(w, \epsilon)|$, $M = \bar{\mathcal{F}}(w, \epsilon, G)$. Recall that these functions are defined in terms of the packing numbers of $F$, so that $N = |G| \Pi(r_w)$ and $M = \Pi(r_{|G|^{1/\nu} w})$. Pick $\delta_w > 0$. For all small enough $w$, we have that:

$$M \geq \Pi(|G|^{1/\nu} (1 + \delta_w) r_w) \geq c_1 |G|^{\frac{d-1}{\nu}} (1 + \delta_w)^{d-1} \Pi(r_w) = c_3 |G|^{1 + \frac{d-1}{\nu}} N \tag{121}$$

where the first inequality comes from equation 114 and the fact that the packing number is a decreasing function of the radius, the second inequality comes from equation 119 and $c_3$ is a constant which depends on the ambient dimension and the curvature of the space $\mathcal{X}$.

Following the arguments of section A.3.2 we see, that by construction, $L(r_w)$ admits only orbits of size $|G|$, end hence in Equation 110, the sum is over all elements in $\bar{\mathcal{F}}_{\mathcal{X}}(r_w, \epsilon, G)$. Hence we recover the same lower bound on sample complexity:

$$T = \Omega\left(\frac{1}{|G|^{\frac{\nu+d-1}{\nu}}} \frac{\sigma^2}{\epsilon^2} \left(\frac{B}{\epsilon}\right)^{\frac{d-1}{\nu}} \log \frac{1}{\delta}\right), \tag{122}$$

This concludes the proof of Theorem 2.

The key element is the original construction of the functions $f_{p,w,\epsilon}(x)$, which are of the form $g(x-p)$ for some bump function constructed around the origin which *is itself symmetric* with respect to the action of $O(d)$ (and hence its subgroups).

# B  Experimental details

## B.1  Synthetic experiments

We use an isotropic Matérn-$5/2$ kernel as the base kernel $k$, and compute $k_G$ according to Equation (5). To generate the objective functions, we first generate a large number $n$ of sample pairs $(x_i, f(x_i))$ from a zero-mean GP prior with kernel lengthscale $l = 0.12$. We scale $n$ with the dimensionality of the problem $d$, so that $n = 64$ for $d = 2$, $n = 256$ for $d = 3$, and $n = 512$ for $d = 6$, to mitigate sparsity. We then use BoTorch to fit the *noise parameter* of a noisy invariant GP with fixed lengthscale $l = 0.12$ to that data, and treat the mean of the trained GP as our objective function. In this way, we ensure that the objective function belongs to the desired RKHS.

During the BO loop, the learner was provided the true lengthscale, magnitude, and noise variance of the trained kernel, to avoid the impact of online hyperparameter learning. For UCB, we use $\beta = 2.0$, except in the quasi-invariant case with $\varepsilon = 0.05$ where we found that increasing $\beta$ was necessary to achieve good performance ($\beta = 3.0$). MVR has no algorithm hyperparameters.

For the constrained BO baseline, we replaced BoTorch's default initialisation heuristic with rejection sampling in order to generate multiple random restarts that lie within the fundamental domain. We found that this improved the diversity of the initial conditions, and hence boosted the performance of the acquisition function optimiser.

In all other cases BoTorch defaults were used.

## B.2  Fusion experiment

We consider the 3-block permutation group, with $|G| = 4! = 24$. The kernel hyperparameters are learned offline, based on 256 runs of the fusion simulator with parameters generated by low-discrepancy Sobol sampling. Due to the cost of evaluating the objective function we use a batched version of the acquisition function as implemented by BoTorch, querying points in batches of 10. We use the multi-start optimisation routine from BoTorch for finding the maximum acquisition function value.

