# OpenReview forum: "Sample-efficient Bayesian Optimisation Using Known Invariances"
_NeurIPS.cc/2024/Conference — NeurIPS 2024 poster_

### Official Review · Reviewer_eKC1 · 2024-06-19

**Soundness:** 3
**Presentation:** 3
**Contribution:** 2
**Rating:** 6
**Confidence:** 2

**Summary:**

This paper provides sample complexity bounds for Bayesian optimization under settings with invariant kernels. Invariant kernels are able to model functions which are invariant under transformation families. Such models allow us to carry out Bayesian optimization efficiently due to having to being able to obtain far much information with a single query than if we used standard kernels (as we obtain information over the whole group of invariances with a single observations, while standard BayesOpt will carry out redundant sampling). The theoretical effect of invariances is investigated in this paper.

A simultaneously invariant kernel satisfies the property that $k(\sigma(x), \sigma(y)) = k(x, y) \forall \sigma \in G$. Where $G$ is the group of invariance. The paper focuses on finite subgroups of isometries, which maintain distances and dot products and therefore any stationary kernel will be simultaneously invariant for any group in question. The paper shows in Proposition 1 how a simultaneously invariant kernel can be used to build an RKHS with the invariances translated to the function space as well. We can then use this space of functions with their corresponding kernel to model the black-box functions for Bayesian Optimization.

The analysis is carried out for two standard BayesOpt algorithms: (a) maximum variance reduction, which selects as the next query the point with maximal variance, and (b) the standard upper confidence bound acquisition function. The first main result is shown in Theorem 1, where the authors obtain an upper bound on the maximal information gain of any bandit algorithm, and of particular importance is a factor of $O(1 / |G|)$. They then show this translates to a $O(1 / |G|)^{(2v + d - 1) / 2v}$ dependency on the sample complexity of the maximum variance reduction algorithm under a Matern kernel. A lower bound is then shown, with complexity $O(1 / |G|)^{(2v + d - 1) / v}$. This shows the upper bound is relatively tight, and that algorithms are likely to fail when run for anything lower than the sample complexity.

Two main experiments are ran:

1. The first experiment looks at the optimization of Matern function samples under different invariance groups. This results in significant speed-ups for optimization in all three benchmarks. In the PermInv6D benchmark, further analysis is carried out where only certain subgroups of the invariances are included in the known invariance group; these result in the expected outcome and the regret improves as more invariances are added.

2. A real-world example is then introduced, where a simulated nuclear reactor configuration is optimized. A subgroup of the invariances is used for computational efficiency, and the results show a very significant improvement from using invariant kernels.

**Strengths:**

- (Originality and Significance) I am not too familiar with the literature around regret bounds in BayesOpt, however, the regret bounds provided in the paper appear to be novel and of significance to the Bayesian optimization community. They can help understand the trade-offs when choosing invariances for kernels. In addition, the empirical study highlights some important properties of using invariances in kernels which are of practical importance in potentially many applications, as highlighted by the nuclear fusion example.

- (Quality) The ideas in the paper seem sound, and follow natural intuition. The method is described and investigated in depth through the theoretical contributions.

- (Clarity) The paper is very well written, and the ideas are expressed clearly. I found it easy to follow, even though I am not well acquainted with the literature around regret bounds.

**Weaknesses:**

- The modeling comes from previous literature, and the bandit strategies followed are standard. The paper's contributions are strictly the theoretical additions, and the convincing, yet somewhat short, empirical study.

- From my understanding, sample complexity upper bounds are only given for the maximum variance reduction algorithm, which could be argued is less important for many optimization applications than UCB.

- Perhaps more applications of the algorithm could be showcased or discussed.

**Questions:**

- How do you do the batching of the acquisition function in the nuclear fusion experiment?

- Is there anything that can be said about the sample complexity of the UCB acquisition function? From L195 I am under the impression that the upper bound only holds for MVR.

- Is there more intuition into why the bounds for maximal information gain are independent of the bandit strategy chosen?

**Limitations:**

Limitations are addressed in Section 5.

---

> ### Author Rebuttal · Authors · 2024-08-06
>
> We thank the reviewer for the close attention paid to our methods and results and for the relevant comments. Below we provide answers to the questions and the remark on the weaknesses identified.
>
> ### Response to weaknesses and questions
> > "The modeling comes from previous literature, and the bandit strategies followed are standard. The paper's contributions are strictly the theoretical additions, and the convincing, yet somewhat short, empirical study. [...] Perhaps more applications of the algorithm could be showcased or discussed."
>
> We would like to suggest that the fact that we rely on previous literature for the core definitions and prerequisite results is not a weakness in of itself. We take great care not to overclaim our contributions, but believe they make sufficient advancement into the field of BO and learning with symmetries, that they present a benchmark of interest to the community. With respect to the empirical study, while we cannot extend our fusion study with further settings at this time, we have chosen to **extend the range of examples and provide other synthetic baselines**, as requested by other reviewers as well. **Please see the attached PDF.**
>
> > "From my understanding, sample complexity upper bounds are only given for the maximum variance reduction algorithm, which could be argued is less important for many optimization applications than UCB. [...] Is there anything that can be said about the sample complexity of the UCB acquisition function? From L195 I am under the impression that the upper bound only holds for MVR."
>
> The sample complexity of UCB is a topic of active research in the frequentist setting (where the target function lives in the RKHS) of BO. Only recently have state-of-the-art (SOTA) upper bounds on regret have been established, e.g. $O (T^\frac{\nu+2d}{2\nu +2d})$ ([1]) which are not order optimal. The motivation for selecting MVR is that, in this setting at least one can show order-optimal upper bounds which align to the *a priori* (algorithm independent) ones of Scarlett et. al. (reference 31 in our text). We agree that a thorough analysis of UCB making use of the SOTA methodology would be of interest.
>
> > "Is there more intuition into why the bounds for maximal information gain are independent of the bandit strategy chosen?"
>
> This is an artefact of the definition of maximal information gain, and holds for all acquisition functions, independent of symmetry. In particular, UCB, MVR, TS, MEI (maximal expected improvement), all benefit from the same definition of $\gamma_T$ (see e.g. reference 33 in our text).  The information gain depends on the choice of samples (not the sampling strategy, but merely its output at time $T$), whereas the _maximal_ information gain is a supremum over all sets of samples and is this a function of the kernel and $T$ alone.
>
> > "How do you do the batching of the acquisition function in the nuclear fusion experiment?"
>
> We use the default quasi-Monte Carlo method from BoTorch for batching (see **[2]** and the BoTorch documentation for details).
> As this is a very standard tool and technique, we did not deem it necessary to comment further. **We will, however, update the relevant sentence to read:**
> > ... we use a quasi Monte Carlo batched version of UCB from BoTorch [X],...
>
> ### Summary
>
> We feel we have addressed the reviewer's main concerns, and would be eager to engage in further discussion. If any of our answers served to clarify and remove the reviewer's doubts, we would be grateful to receive an even firmer acknowledgement of our paper from the reviewer.
>
>
> **References:**
>
> [1] On the Sublinear Regret of GP-UCB, Justin Whitehouse et. al.
>
> [2] BoTorch: A Framework for Efficient Monte-Carlo Bayesian Optimization; Balandat et al. (2020)

---

> > ### Comment · Reviewer_eKC1 · 2024-08-07
> >
> > Thanks for the response, the clarifications, and for the additional experiments.
> >
> > > ...but believe they make sufficient advancement into the field of BO and learning with symmetries, that they present a benchmark of interest to the community
> >
> > I do agree with this, which is why I recommended acceptance. However, as the contributions are mainly theoretical, and I am not well acquainted with the theoretical side of BO, it is difficult for me to measure the strengths of the paper (which is the reason for my low confidence).
> >
> > Regarding the difficulty to come up with bounds for the UCB algorithm, it remains a weakness of the paper even if a difficult task.
> >
> > Based on these, I will stand by my original review; I liked the paper and I recommend acceptance, however, with low confidence.

---

### Official Review · Reviewer_pMXJ · 2024-07-07

**Soundness:** 3
**Presentation:** 3
**Contribution:** 3
**Rating:** 6
**Confidence:** 3

**Summary:**

The authors proposed a new setting where the invariance is either known or partially known. They theoretically examined the upper and lower bounds of convergence rates concerning sample complexity. Their findings demonstrated both theoretical and empirical superior performance over the standard UCB approach in cases with known invariance. Additionally, they explored more practical settings with partially known invariance using synthetic tasks and a real-world example from tokamak optimization.

**Strengths:**

- The theoretical section is well-written, but the experimental part is difficult to follow.
- The convergence rate is provably better than vanilla UCB and has shown empirical success in both synthetic tasks and one real-world example.

**Weaknesses:**

**Limited Applicability:**
I am not convinced that this new setting is widely applicable to real-world scenarios. Aside from the specific tokamak application, it is challenging to identify other examples where this setting would be useful. While regression tasks in previous work have broad applicability, is the same true for optimization? Can you provide more intuitive examples of invariance-aware BO? For instance, in classification tasks, such as translation invariance in cat images in the introduction section, the concept is clear. However, does this example apply to BO? Beyond the tokamak example, the authors mention material science, but it is difficult to envision practical situations in this domain where the search space has known invariance. For example, while crystal lattice invariance is known, is this relevant to the search space? This is merely one feature characterizing materials, and it is hard to imagine a scenario where we aim to maximize something within this symmetric lattice space. While band gap optimization can be the one (finding minimum and maximum of electronic energy), it is computationally simple and not computationally demanding.

**Feasibility of Listing Possible Invariances:**
It is hard to imagine users being able to list possible invariances. This raises concerns about the motivation behind the work.

**Lack of Simpler Baselines:**
The work lacks comparisons with simpler baselines, such as periodic kernels or constrained BO. For example, in Figure 2, if the search space is known to be symmetric for 10 cycles, why not constrain the search space to 1/10 as in typical constrained BO approaches? Finding all peaks in Figure 2 seems wasteful since we know they are repetitive. The same applies to permutation invariance; why not algorithmically reject repetitive candidates?

**Clarity in Experimental Section:**
The experimental section is difficult to follow. The tasks in Figure 3 are unclear, making it hard for readers to reproduce the results.

**Minor Points**
- Numerous typos (e.g., lines 102-103, 116-117, repeating the RKHS acronym definition).
- Limited baseline comparisons (e.g., permutation kernel, periodic kernel, and constrained BO).

**Questions:**

The questions are detailed in the weakness section above.

**Limitations:**

Limitations are well discussed.

---

> ### Author Rebuttal · Authors · 2024-08-06
>
> We thank the reviewer for their comments, and for their close reading that brought to our attention minor typographical errors.
>
> ## Response to weaknesses
> ### Limited applicability
> We respectfully disagree with the reviewer's comment on the limited applicability of our work.
> Some examples:
> - Molecular optimization, where the invariance is in the permutation of the molecule as a list of atoms / functional groups **[1]**.
> - Cache memory placement in computer architectures. In a fully associative two-level cache with LRU eviction, the permutation of function symbols in tightly coupled memory (TCM) is invariant and does not impact hard real-time (HRT) performance. This invariance is well-known to system designers **[2]**.
> - Compiler optimization operates on intermediate representations (IR) of code and performs passes to create the most efficient IR for back-end translation. Some of these passes operate on the IR through permutation of basic blocks only, rearranging the state in the hope that further passes will benefit **[3]**.
> - Placement and floorplanning is a 2D graph embedding problem where a graph of components must be embedded in 2D under a series of geometric constraints with geometric symmetries in the arrangement (i.e. 4-fold rotations and 2 fold reflections) **[4]**.
>
> ### Feasibility of listing invariances
> We don’t believe this is a flaw in our approach, as seen in the examples above. Moreover, in Figure 3 (last column), we demonstrate that having partial knowledge of the invariances in the system can still lead to significant gains.
>
> ### Clarity in the experimental section
> We would appreciate it if the reviewer could specify which paragraph is unclear so that we can improve the exposition. By default **we will modify the opening paragraph to read**:
>
> > For each of our synthetic experiments, our objective function is drawn from an RKHS with a group-invariant isotropic Matern-5/2 kernel. To generate the function, we first sample $n$ points from a GP prior with the target kernel. Then, we fit a GP to those samples and use the posterior mean function of the fitted GP as the objective function. The groups we consider act by permutation of the coordinate axes, and therefore include reflections and discrete rotations. For *PermInv-2D* and *PermInv-6D*, the kernel of the GP is invariant to the full permutation group which acts on $R^2$ and $R^6$ respectively by permuting the coordinates. For *CyclInv-3D* it is invariant to the cyclic group acting on $R^3$ by permuting the axes cyclically. The observations of the objective values are corrupted with Gaussian noise of known variance. In our examples, we provide the learner with the true kernel and noise variance to eliminate the effect of online hyperparameter learning. The values of the GP hyperparameters are provided in the appendix. The objective functions *PermInv-2D* and *CyclInv-3D* are visualised in Figures 1a and 1b respectively.
>
> ### Lack of simpler baselines
> We appreciate that the reviewer has considered the setting carefully, and agree that baselines like constrained BO (CBO) make sense to investigate.
> We also hope that as the reviewer has included this section in "minor points", it will be sufficient to improve this section of the manuscript by **adding a comparison section which we summarise below (see the PDF for appropriate figures)**.
>
> - CBO vs. ours
>     - While constraining the domain to the fundamental region of the action is a valid choice of experimental design, characterizing the region analytically can be difficult when the dimension of the domain is high and the group is complicated. Moreover, optimizing an acquisition function using global optimizers in such complex input spaces is non-trivial.
> Constraining by rejection sampling may lead to inefficient use of computational resources and in practice could significantly slow down acquisition function optimisation. In contrast, our approach is both efficient and elegant in handling these challenges.
>     - In terms of sample complexity, CBO is expected to achieve only a $\frac{1}{|G|}$ reduction. The intuition is that in CBO the kernel is not aware of all of the added structure that the RKHS elements possess, i.e. while we have restricted the _domain of exploration_, we have not restricted the _function class_. We will add a theoretical justification in the appendix.
>
> - Other methods
>     - For periodic kernels, our knowledge is that they are used to model data repeating at regular intervals, such as time-series data or spatial patterns with cyclic behavior.
> The former are out of scope for this paper, as we deal with finite groups of transformations (discrete groups on compact domains), which are not applicable to time-series.
> The latter represent a subset of our problem, in which case a "periodic" kernel is the "invariant" kernel with respect to the cyclic group.
>
> ## Summary
> We would like to thank the reviewer again for engaging with our work.
> We hope, however, that the reviewer would consider re-evaluating the paper for its intended purpose, i.e. mainly making a theoretical contribution rather than an empirical one.
> We have addressed all the questions raised in the review, with a particular focus towards the empirical comparisons, explanations and improvements proposed by the reviewer.
> Based on our changes, we hope that the reviewer would consider that the empirical part of the paper has improved as well.
>
> We kindly request that the reviewer reconsider the decision in light of our responses and update their score accordingly.
> We are also eager to address any additional concerns.
>
> **References:**
>
> [1] Learning Invariant Representations of Molecules for Atomization Energy Prediction; Montavon et al. (2012)
>
> [2] Optimal Data Placement for Heterogeneous Cache, Memory, and Storage Systems; Zhang et al. (2020)
>
> [3] Engineering a Compiler; Cooper et al. (2011)
>
> [4] Algorithms for VLSI Design Automation; Gerez (1999)

---

> > ### Comment · Reviewer_pMXJ · 2024-08-13
> >
> > Thank you for the author for their effort and clear rebuttal. My concerns are adequately addressed, so I raise the score.

---

### Official Review · Reviewer_5kTb · 2024-07-12

**Soundness:** 3
**Presentation:** 3
**Contribution:** 2
**Rating:** 6
**Confidence:** 3

**Summary:**

The paper introduces an Bayesian Optimisation method which is able to take into account known invariances of the objective function.  Specifically, it is assumed that the objective function remains invariant under a finite group action $G$.

The approach is straightforward, one does standard Bayesian optimisation using a $G$-invariant kernel, following the approach of [Haasdonk et al, 2007] and others.    The authors then propose using standard BO with UCB or Maximum variance acquisition function.
  The paper provides theoretical bounds on the regret for this invariance aware approach -- in the special case where the domain is a sphere.    The authors demonstrate the sample efficiency of the method empirically, on some synthetic experiments, and then a nice tokamak optimisation example.

**Strengths:**

The specific approach of using a G-invariant kernel in the context of Bayesian optimisation seems new to me.

The biggest contribution of the paper is the very nice sample complexity bounds,  both lower and upper -- albeit in a simplified setting, but really demonstrate the mechanisms from which the efficiency gain arises in this setting.

 The tokamak numerical experiment is quite interesting and a challenging example.

**Weaknesses:**

Dealing with invariances in optimisation (Bayesian / black-box / otherwise) is certainly not a new problem, and there are typically well-established methods to deal with this, through symmetry breaking constraints etc.   This is particularly common in the context of computational chemistry and materials design properties where molecules have a wide range of symmetries.   One noteworthy example is Bayesian Optimization With Symmetry Relaxation Algorithm from [Zuo, Yunxing, et al. "Accelerating materials discovery with Bayesian optimization and graph deep learning." Materials Today 51 (2021): 126-135].     So while the approach is novel, it is far from unique, and I feel that the authors have not really engaged with existing methods at all.   The literature review seems to go from "Invariances in DL" -> "Invariant Kernels" -> "Kernels on Manifolds" (again missing a lot of important works there too) -> "regret bounds for BO".

The sample complexity bounds are certainly novel, but for example the upper bound largely seems to follow Joan Bruna's existing work on this which tackles a very similar problem.

So to summarise:
(+) I think this is nice work.
(+) I think its a novel approach.
(+) The theory is nice.
(-) However, it feels like a quite incremental contribution, which makes no attempt to engage with alternative approaches to the same problem.
(-) The theory, while nice, again seems a moderate modification of existing work.

**Questions:**

1. Could the authors help distinguish their work from other approaches in this field.   I strongly urge the authors to engage a bit more with other proposed methodology (some of which might not be very rigorous).  Understandably this is a fairly common issue, and many resolutions exist -- certainly not all as nice as this paper.

2. In particular, it would be good if the authors could identify strong advantages of this method over others, and maybe even provide other methods as a useful benchmark.

3. One possible advantage of this approach over others is situations where the invariance is not strongly constrained, e.g. quasi-invariances.  Could the authors demonstrate that this method still yields better sample complexity when the objective is nearly invariant?

I am v happy to adjust my score of the authors can provide a stronger case for the novelty and/or value-add of this work over other approaches.

**Limitations:**

The limitations addressed are around the complexity of computing the invariant kernel -- this has been addressed and some guidelines for mitigating this are discussed.

---

> ### Author Rebuttal · Authors · 2024-08-06
>
> We thank the reviewer for their engagement with our paper and recognition of its "novel approach" to a popular problem.
>
> ## Response to weaknesses
> ### Critique of literature review
> We cannot hope to provide an exhaustive survey of the literature in this setting.
> We acknowledge, however, that the reviewer is right in requesting a broader coverage of the field in the literature review. **We will include the following paragraph to engage with the literature on empirical methods.**
>
> > **Physics informed BO.** There is an extensive body of literature where symmetries come from physical information present in the system states, which is well known to the experiment designer. Our own example (Figure 4) benefits from such a description.
> In **[1]** authors apply BO after a variety of physics-informed feature transformations to the state, and engineer product kernels with factors acting on these individual representations.
> In **[2]** the authors employ BO to handle optimization over crystal configurations with symmetries found using **[3]**.
> The objective is a surrogate for the physical target of system energy, computed via a GNN architecture.
> Both the symmetries and the invariance of the objective lack theoretical guarantees, but empirical results show that constraining the symmetry by separating the dependent and independent parameters provides significant gains in terms of regret versus non-constrained ablations.
> Asymptotic rates on sample complexity are not estimated.
> In **[4]** a set-invariant kernel is constructed in the same way as ours (see Eq 38, and references [16, 29] from the original manuscript) to optimize functions over sets, incorporating the permutation invariance of the set elements.
> >
> > **References:**
> >
> > [1] A physics informed Bayesian optimization approach for material design: application to NiTi shape memory alloys;
> Khatamsaz et al. (2023)
> >
> > [2] Accelerating Materials Discovery with Bayesian Optimization and Graph Deep Learning; Zuoa et al. (2021)
> >
> > [3] Spglib: a software library for crystal symmetry search; Togo et al. (2018)
> >
> > [4] Bayesian optimization with approximate set kernels; Kim et al. (2021)
>
> ### Critique of methods
> We are grateful to the reviewer for the praise of our exposition and novelty of our approach. We respond below to their criticism.
>
> > "...for example the upper bound largely seems to follow Joan Bruna's existing work on this which tackles a very similar problem."
>
> The upper bound is dependent on results involving the ratio of the dimensions of invariant RKHS eigenspaces vs. their non-invariant counterparts. We quote Bruna et al. (reference [3] in our manuscript), for these results; however, it is important to note that their paper is concerned with Kernel Ridge Regression (KRR) alone, which has fundamental differences in both theory and algorithms compared to BO.
> Beyond quoting the above result, our proofs proceed largely independently. A careful computation of how these ratios factor into the bound for maximal information gain is performed. Then we turn the bounds on information gain into a bound on sample complexity.
> The analysis in this case, and for these particular bounds, is, to the best of our knowledge, novel, and so is the derived result.
> Finally, a significant portion of our paper is devoted to finding a lower bound for this setting (analysis which is not present in Bruna et al.), which we feel deserves more acknowledgment.
>
> > "...which makes no attempt to engage with alternative approaches"
>
> As above, we feel this remark is somewhat uncharitable, as the main contribution of the paper is fundamentally a theoretical novelty, not an empirical one. However, we acknowledge that a stronger comparison with existing methods is valuable for members of the research community. *As such we will include additional baselines in our empirical results section, comparing with data augmentation and constrained BO (see the attached PDF and responses to other reviewers).*
>
> > "The theory, while nice, again seems a moderate modification of existing work..."
>
> In our paper, we have taken care to acknowledge papers that follow a similar method or contributed to our thought process.
> Therefore, while we would agree that some of the steps in the proofs may be familiar to a well-versed practitioner, we would like to argue the analysis for the incorporation of *symmetries* in particular is novel and showcases the phenomenon under study (that of sample complexity decrease due to symmetry) well.
>
> ## Answers to questions
> > "Could the authors help distinguish their work from other approaches [...] -- certainly not all as nice as this paper."
>
> Please see the additional discussion of the literature and the PDF. We primarily **provide comparisons of our method with data augmentation and constrained BO.**
>
> > "[...] quasi-invariances. Could the authors demonstrate that this method still yields better sample complexity [...]?"
>
> This is indeed a very pertinent question and one that we do not neglect in our paper, as we present the relative gains in sample complexity by progressively incorporating more knowledge of the underlying symmetry group in Figure 3.
> **In the attached PDF, we demonstrate an example of a quasi-invariant function**, and demonstrate that MVR achieves good performance even as the function becomes less strongly invariant. **We will incorporate this in a new subsection of our experimental results.**
>
> ## Summary
> We have addressed all the questions raised by the reviewer.
> We believe that our theoretical results, combined with our practical experiments, provide strong evidence of the utility of our approach.
> We kindly request that the reviewer reconsider their decision in light of our responses and their comments that this is "nice work" and "a novel approach".
> We are also eager to address any additional concerns the reviewer may have.

---

> > ### Comment · Reviewer_5kTb · 2024-08-12
> > **Response**
> >
> > I thank the authors for their rebuttal and for their suggested updates.   I feel that I understand a bit better the authors contributions on the theoretical aspects of this work, which are the main contribution of this paper, in particular how they distinguish with other works in this area.    I think this is broadly a well written paper, but I do agree with the other reviewers that perhaps some other applications should have been showcased to demonstrate the feasibility of identifying (partial or otherwise) invariance.   I will update my score.

---

### Official Review · Reviewer_8csK · 2024-07-13

**Soundness:** 3
**Presentation:** 3
**Contribution:** 3
**Rating:** 7
**Confidence:** 3

**Summary:**

The paper targets the Bayesian optimization problem for a class of invariant functions, which is useful in many fields including machine learning and physics. Specifically, the paper proposes to incorporate the invariances into the kernel of the GP to produce invariance-aware algorithms, either fully or partially. The paper presents theoretical analysis regarding the lower bound on sample complexity when using invariant kernels in BO. Several experiments are shown to support the findings.

**Strengths:**

•	The target problems (invariant functions) are important yet has not been focused on much.

•	The ideas of invariant kernels are interesting and worth investigating.

•	The theoretical analysis is thorough by providing both upper and lower bound for sample complexity.

•	The empirical performance is good in all cases.

**Weaknesses:**

•	In Figure 3, the plots should be consistent, i.e., either use cumulative regret or simple regret.
•	The experiments are only conducted on low-dim problems.

**Questions:**

I don’t have many questions regarding the submission. I think my main question is on the performance of the algorithms for higher-dimensional problems. Will the methods work well in high-dim regime? And if not, are there anything we can do to make it work better in the high-dim setting?

**Limitations:**

I agree with the author’s limitations regarding the computational expense of fully invariant kernel on large group of transformation.

No negative societal impact needs to be addressed.

---

> ### Author Rebuttal · Authors · 2024-08-06
>
> We thank the reviewer for their appreciation of our work, and their positive comments regarding its importance, thoroughness and empirical performance.
>
> ### Response to weaknesses and questions
>
> **Concerning the plots in Figure 3:**
> Currently, we plot the cumulative regret for UCB and the simple regret for MVR, which we suggest is standard practice for these algorithms.
> In the literature, the regret bounds for UCB are reported in terms of cumulative regret **[1, 2]** and those for MVR in terms of simple regret **[3]**.
> We chose to follow this convention for our bounds and experiments.
> This makes it easy to identify the performance improvement that can be achieved by incorporating invariance compared to existing regret bounds.
> Finally, we would highlight that it is fairly trivial to convert the simple regret to cumulative regret, if that is interesting for the reader.
>
> **Concerning the performance in high dimensions:**
> We agree that the performance of any algorithm in high dimensions is important for real-world applications; however, BO in high dimensions is a domain in its own right and often necessitates special treatment.
> In our work, we do evaluate performance on medium dimensional problems (6 and 12 dimensions), which are already bordering on the point at which traditional BO breaks down.
> Our method does show improved performance with increasing dimension, but beyond 10-15 dimensions it is likely that the factors that limit the performance of standard methods will also start to hamper ours.
>
> On the other hand, our kernel is an additive kernel, which is an established method for improved performance in high dimensions (see, e.g., [4]). With this in mind, **we will add a reference to the literature on additive methods for high-dimensional BO, namely [4] and [5]**.
>
> We wholeheartedly agree with the author that this is certainly an interesting topic, but given the additional scope involved we suggest that investigating methods for high-dimensional invariant functions might be better suited to a dedicated paper.
>
>
> **References:**
>
> [1] Gaussian Process Optimization in the Bandit Setting: No Regret and Experimental Design; Srinivas et al. (2009)
>
> [2] On Information Gain and Regret Bounds in Gaussian Process Bandits; Vakili et al. (2021)
>
> [3] Optimal order simple regret for Gaussian process bandits; Vakili et al. (2021)
>
> [4] High-Dimensional Bayesian Optimisation and Bandits via Additive Models; Kandasamy et al. (2016)
>
> [5] High-Dimensional Bayesian Optimization via Additive Models with Overlapping Groups; Rolland et al. (2018)

---

> > ### Comment · Reviewer_8csK · 2024-08-14
> >
> > Thank you for the response. The response addresses my concerns so I keep my current score.

---

### Author Rebuttal · Authors · 2024-08-07

# Global rebuttal

The authors thank the reviewers collectively for engaging with the work, and providing both positive feedback and actionable critique of our manuscript.
We believe we have provided satisfactory answers, and produced a further body of evidence that strengthens the cause for our paper.
The following is a (reductive) summary of the 4 reviewer's concerns, and the work we have done to address them (highlighted in **bold**).

## Outline of main reviewer concerns
- Comments were made about the breadth and scope of our empirical results.
- Comments were made about the literature review, requesting us to engage with a broader body of literature (especially applied works that may have employed similar methods).
- A request for clarification was made about our experimental section.

## Outline of our actions
- In the attached PDF, we have added **details of several additional experiments**:
  - Figure 1 details a **new example of quasi-invariant optimisation**.
  - Figure 2 adds **comparisons with constrained BO** to our regret plots.
  - Figure 3 adds **comparisons with data augmentation** to our performance plots.
- We have **extended the scope** of our literature review.

### Quasi-invariance
Reviewer 5kTB requested an exploration of quasi-invariance. The notion of 'almost'-invariance has been mentioned in previous literature, but has not been considered in BO. However, one of the strengths of the invariant kernel method is the ease of applicability to the quasi-invariant setting, by considering a kernel that is comprised of a sum of invariant and non-invariant components.

In response to the reviewer's comment, we have **included plots showing the performance of our algorithm in this setting**, comparing the performance of a non-invariant kernel, an invariant kernel, and the aforementioned sum kernel. The plots show that as long as the objective does not deviate too much from being invariant, the invariant kernel significantly surpasses the non-invariant kernel and is comparable in performance to the sum kernel.

### Constrained BO and data augmentation
Reviewer pMXJ requested more baseline comparisons. In response, we have **added plots to compare our method against constrained BO and data augmentation**. In doing so, we have also responded to reviewer eKC1's request to extend the scope of our empirical study.

We show that using data augmentation leads to exploding memory requirements while our method remains effective and lightweight.

We use the built-in functionality of BoTorch to implement a constrained BO benchmark on our test problems, ensuring that the acquisition function optimisation step remains as similar as possible to the unconstrained case. We are happy to discuss more details of our implementation if the reviewer is interested. We would like to highlight two key takeaways from this new benchmark:
1. As this method requires hand-writing an analytic description of the the fundamental domain's boundary, setting up the constrained BO problem becomes very difficult for groups with a more complicated action. Although we were able to implement it for the full permutation group, this is a specific example where the fundamental region can be computed with ease; it is not as straightforward to implement even for subgroups of the permutation group (e.g. the cyclic group from our experimental section).
2. Our invariant kernel method significantly outperforms constrained BO. In the reply to reviewer pMXJ and the caption of Figure 2 in the PDF, we have provided intuition as to why it is expected that constrained BO achieves worse sample complexity than our method. **We will add a detailed proof of this fact in the appendix.**

### Literature review and scope
To address reviewer 5kTB's concerns about engaging with previous literature, we will **add the given paragraph and references to the literature review** (see reply to 5kTB) concerning existing methods for incorporating structure into BO. We gratefully acknowledge the reviewer's contribution in encouraging us to include this section, as we believe it strengthens our paper. Nonetheless, we would like to remind the reviewers that many of these methods are purely empirical, whereas our work is primarily concerned with providing an algorithm with performance guarantees that are grounded in theory.

We will also **add references to high-dimensional BO**, following comments from reviewer 8csK.

We hope that our additional benchmarks and new quasi-invariance experiment, alongside the additional examples we list in our reply, will address reviewer pMXJ's question about the limited applicability of our methods.

### Experimental section
To add clarity to the experimental section, we've included a paragraph with further details and explanations at reviewer pMXJ's request.

We will also include the hyperparameters of the GPs used in a table in the appendix.

---

### Author Response · Authors · 2024-08-12
**Request for response**

We thank all the reviewers again for their time taken to review our work.

**As the author-reviewer discussion period is nearing its end, we would appreciate it if the reviewers would respond to our rebuttals.**

If there any further questions, we are more than happy to clarify and discuss further.

---

### Author Response · Authors · 2024-08-13
**Rebuttal summary and acknowledgements**

To all reviewers and area chairs,

As the rebuttal period it nearing its end, we would like to thank all the reviewers again for taking the time to read our manuscript, to provide constructive criticism and raise many important points that have helped improve both our understanding and the write-up.

We would like to doubly thank reviewers for engaging in the rebuttal discussion and for providing _unanimous approval for our paper_. We will improve the manuscript according to the discussion below to ensure that it passes the high quality standards expected from a NeurIPS publication.

---

### Decision · Program_Chairs · 2024-09-25

**Decision:**

Accept (poster)

**Comment:**

The reviewers discuss the incorporation of invariance under a group action into a standard UCB-based BO loop, and show that this leads to improvements both empirically and theoretically.  Coming out from the discussion period the reviewers were unanimously positive about the paper.  This seems like a clear case for acceptance.